# Parameter-Efficient Fine-Tuning of State Space Models

Kevin Galim [*1]  Wonjun Kang [*1 2]  Yuchen Zeng [*3]  Hyung Il Koo [1]  Kangwook Lee [3]

## Abstract

Deep State Space Models (SSMs), such as Mamba (Gu & Dao, 2024), have become powerful tools for language modeling, offering high performance and linear scalability with sequence length. However, the application of parameter-efficient fine-tuning (PEFT) methods to SSM-based models remains largely underexplored. We start by investigating two fundamental questions on existing PEFT methods: (i) How do they perform on SSM-based models? (ii) Which parameters should they target for optimal results? Our analysis shows that LoRA and its variants consistently outperform all other PEFT methods. While LoRA is effective for linear projection matrices, it fails on SSM modules—yet still outperforms other methods applicable to SSMs, indicating their limitations. This underscores the need for a specialized SSM tuning approach. To address this, we propose Sparse Dimension Tuning (SDT), a PEFT method tailored for SSM modules. Combining SDT for SSMs with LoRA for linear projection matrices, we achieve state-of-the-art performance across extensive experiments.

## 1. Introduction

In the past few years, Large Language Models (LLMs) such as ChatGPT (Achiam et al., 2023; Brown et al., 2020) have achieved groundbreaking performance and are now widely used in daily life. While many models rely on the Transformer architecture (Vaswani et al., 2017), its quadratic time complexity due to the attention mechanism poses challenges for long sequences. To address this, alternative architectures such as linear attention (Katharopoulos et al., 2020), RWKV (Peng et al., 2023), RetNet (Sun et al., 2023), and Mamba (Gu & Dao, 2024) have been developed, offering

subquadratic time complexity. Efficient attention alternatives often rely on State Space Models (SSMs) or their variants (Gu et al., 2021; 2022b;a; Gu & Dao, 2024), which are akin to linear RNNs, maintaining hidden states of fixed size for sequential processing. S4 (Gu et al., 2022b;a) overcomes RNNs' parallel training limitations by constraining parameter structures, enabling a convolutional form for efficient parallel computation. S6 (Gu & Dao, 2024) improves this with input-dependent parameters, enabling selective focus on relevant information per token. Building on S6 with linear projection matrices (analogous to the Feed-Forward Networks in Transformer layers), Mamba-I (Gu & Dao, 2024) emerged as a prominent SSM-based model. Mamba-I was later extended to Mamba-II (Dao & Gu, 2024), with both models achieving Transformer-level performance in language modeling and gaining widespread recognition.

As SSMs gain popularity, performing parameter-efficient fine-tuning (PEFT) on pretrained models for downstream tasks is crucial, since full fine-tuning is costly and inefficient. Numerous PEFT methods (Houlsby et al., 2019; Hu et al., 2021; He et al., 2021; Li & Liang, 2021; Lester et al., 2021; Zaken et al., 2022; Liu et al., 2021; 2022; Houlsby et al., 2019) have been developed, achieving notable success on Transformer models. The most popular PEFT methods fall into three categories: (i) *input-injection methods*, which add sequences to the model's main input (Lester et al., 2021) or prepend tokens to the intermediate inputs at each layer (Li & Liang, 2021); (ii) *architecture-enhancement methods*, which adjust the model architecture. For example, Houlsby et al. (2019) added layers between Transformer layers, while Additional-scan (Yoshimura et al., 2025) expands state dimensions in the SSM module; (iii) *weight-tuning methods*, which directly modify existing model weights. Notable weight-tuning approaches include BitFit (Zaken et al., 2022), which updates only bias terms, and LoRA (Hu et al., 2021), which modifies weight matrices through low-rank updates, along with its variants such as DoRA (Liu et al., 2024) and LoRA+ (Hayou et al., 2024). For simplicity, we denote LoRA and its variants as LoRA$^\star$.

Despite the success that existing PEFT methods have achieved in adapting Transformer-based models, their efficacy in adapting SSM-based models remains largely underexplored, leaving many interesting questions open.

---

[*]Equal contribution. Authors listed in alphabetical order. [1]FuriosaAI [2]Seoul National University [3]University of Wisconsin-Madison. Correspondence to: Kangwook Lee <kangwook.lee@wisc.edu>.

*Proceedings of the $42^{nd}$ International Conference on Machine Learning*, Vancouver, Canada. PMLR 267, 2025. Copyright 2025 by the author(s).

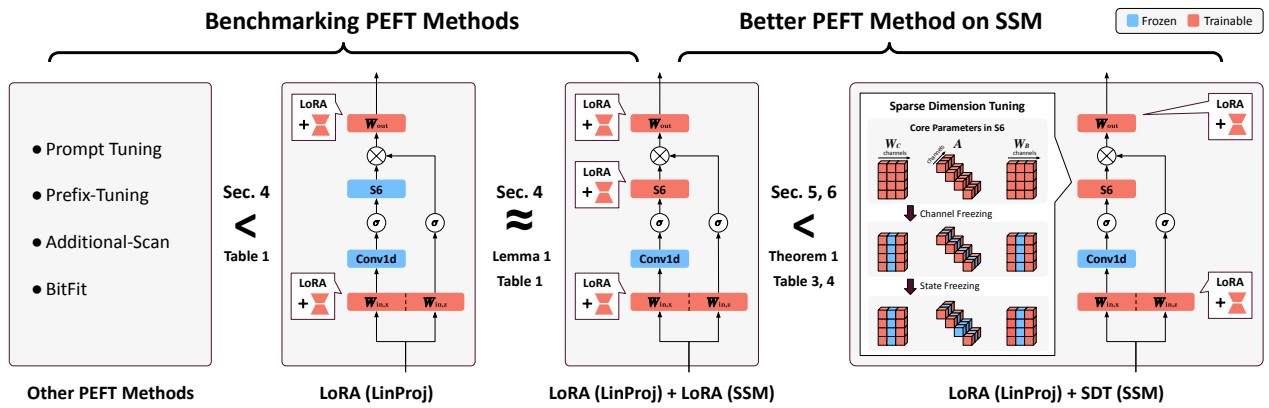

*Figure 1.* **A visual guide to PEFT methods in SSM-based models: benchmarking and innovation.** We compare various existing PEFT approaches on SSM-based models, demonstrating that LoRA applied to linear projection matrices outperforms all other methods. However, extending LoRA to SSM modules fails to yield further improvements. To address this, we propose Sparse Dimension Tuning (SDT), which achieves state-of-the-art performance on SSM-based models when combined with LoRA for linear projection matrices.

1. *Do existing popular PEFT methods remain effective for SSM-based models?*
2. *If applicable, what is the optimal way to integrate these methods into SSM-based models, and which parameters should be updated?*
3. *If not, can we design specialized variants tailored to SSMs that yield superior performance?*

Our main contributions to address these questions are:

- **Comprehensive Benchmarking of PEFT Methods.** We benchmark six widely used PEFT methods across three categories on diverse tasks, including natural language understanding, generation, and computer vision. We evaluate these methods on both SSM-based models (i.e., Mamba) and a hybrid model (i.e., Jamba (Lieber et al., 2025)), which consists of both Transformer layers and Mamba layers. Our results show that LoRA$^\star$ consistently outperforms all other PEFT methods on both SSM-based and hybrid models. However, its effectiveness is limited to linear projection matrices, as further tuning of SSM modules does not improve performance. Notably, other methods applicable to SSM modules perform worse than LoRA$^\star$, further underscoring the need for a specialized approach to tuning SSM modules.

- **Introducing Sparse Dimension Tuning (SDT) for SSM Modules.** To develop an effective method for tuning SSM modules, we conduct a theoretical analysis to understand the roles of different parameters. This analysis motivates the *Sparse Dimension Tuning and Pruning* (SDT-P) method, which improves efficiency by freezing and pruning certain channel and state dimensions while training only the remaining ones. We establish theoretical guarantees for its effectiveness in SSM-based models when combined with LoRA applied to linear projection matrices. We then simplify SDT-P into *Sparse Dimension Tuning* (SDT) by omitting explicit pruning, as pruned dimensions

can be considered equivalent to training dimensions set to zero. SDT selectively updates channels and fine-tunes specific dimensions within them, as illustrated in Fig. 1.

- **Demonstrating Effectiveness of SDT.** Through extensive experiments, we demonstrate that integrating SDT into SSM-based models, combined with applying LoRA$^\star$ to their linear projection matrices, achieves state-of-the-art fine-tuning performance.

The roadmap of our paper is illustrated in Fig. 1. Our code is available at https://github.com/furiosa-ai/ssm-peft.

## 2. Related Works

**Concurrent Works of PEFT on SSMs.** Several concurrent studies (Halloran et al., 2024; Yoshimura et al., 2025; Kang et al., 2025) have investigated PEFT methods for SSM-based models. Halloran et al. (2024) studied both in-context learning and parameter-efficient fine-tuning, with an orthogonal focus on analyzing Mamba's stability under mixed-precision training using Lyapunov exponents. Kang et al. (2025) introduced state-based PEFT methods and proposed State-offset Tuning, solely focusing fine-tuning Mamba's S6 blocks. Yoshimura et al. (2025) benchmarked multiple PEFT approaches—including established methods and a new method called Additional-scan (which adds a trainable state dimension to the SSM module), plus partial tuning (fine-tuning only a subset of parameters)—and introduced MambaPEFT through PEFT search strategies. While Yoshimura et al. (2025) solely focused on Mamba-I, providing an in-depth study of that particular architecture, our work investigates a broader class of SSM-based models including deep S4, Mamba-I, Jamba in the main body, as well as Mamba-II presented in Sec. C.2 and E.2, aiming to offer general insights on how to effectively tune SSMs rather than focusing on a single variant.

**Sparse Tuning.** Several studies have explored sparse parameter selection in fine-tuning (Song et al., 2024) and skill localization (Panigrahi et al., 2023). Song et al. (2024) showed that sparse tuning is an effective PEFT method, linking the low intrinsic dimensionality of pre-trained models to the proportion of parameters needing updates. They propose selecting optimal fine-tuning parameters based on gradient magnitudes. We enable sparse tuning for SSM by applying sparsity across entire dimensions (channel and state) rather than specific neurons. Panigrahi et al. (2023) focused on identifying neurons responsible for specific downstream tasks by fully fine-tuning the model and computing neuron masks to minimize task loss. While effective for skill localization, this method is computationally expensive and not optimized for parameter-efficient fine-tuning.

In Sec. A, we provide a more detailed discussion of related work on SSMs and PEFT.

## 3. Preliminaries

### 3.1. State Space Models

**Discrete-Time SSMs.** The initial SSM is derived from a specific continuous system that maps a one-dimensional function or signal $x(t) \in \mathbb{R}$ to $y(t) \in \mathbb{R}$ via an $H$-dimensional latent state $\boldsymbol{h}(t) \in \mathbb{R}^H$, as described in (1). In this formulation, input transition vector $\boldsymbol{B} \in \mathbb{R}^{H \times 1}$ indicates the input's impact on the state of the system, state matrix $\boldsymbol{A} \in \mathbb{R}^{H \times H}$ characterizes the system's internal state dynamics, and the output mapping vector $\boldsymbol{C} \in \mathbb{R}^{1 \times H}$ relates the state to the output $y(t)$.[1]

$$
\begin{aligned}
\boldsymbol{h}'(t) &= \boldsymbol{A}\boldsymbol{h}(t) + \boldsymbol{B}x(t) \\
y(t) &= \boldsymbol{C}\boldsymbol{h}(t)
\end{aligned} \quad (1) \qquad
\begin{aligned}
\boldsymbol{h}_t &= \overline{\boldsymbol{A}}\boldsymbol{h}_{t-1} + \overline{\boldsymbol{B}}x_t, \\
y_t &= \boldsymbol{C}\boldsymbol{h}_t
\end{aligned} \quad (2)
$$

$$
\begin{aligned}
\overline{\boldsymbol{K}} &= (\boldsymbol{C}\overline{\boldsymbol{B}}, \boldsymbol{C}\overline{\boldsymbol{A}}\,\overline{\boldsymbol{B}}, \ldots, \boldsymbol{C}\overline{\boldsymbol{A}}^{t-1}\overline{\boldsymbol{B}}), \\
(y_1, \ldots, y_t) &= (x_1, \ldots, x_t) * \overline{\boldsymbol{K}}
\end{aligned} \quad (3)
$$

To handle discrete inputs, the continuous parameters $(\boldsymbol{A}, \boldsymbol{B})$ are discretized into $(\overline{\boldsymbol{A}}, \overline{\boldsymbol{B}})$ using a learnable step size $\Delta \in \mathbb{R}$. A common discretization rule, the zero-order hold, defines $\overline{\boldsymbol{A}} = \exp(\Delta \boldsymbol{A}), \overline{\boldsymbol{B}} = (\Delta \boldsymbol{A})^{-1}(\exp(\Delta \boldsymbol{A}) - \boldsymbol{I}) \cdot \Delta \boldsymbol{B}$. The discrete-time SSM, given in (2), enables efficient inference via long convolution described in (3). For multi-channel inputs $\boldsymbol{x}, \boldsymbol{y} \in \mathbb{R}^D$, separate SSMs are used per channel, with a superscript $(d)$ indicating channel-specific parameters when needed.

**Structured State Space Sequence Model (S4).** S4, introduced by Gu et al. (2022b), is an early application of SSMs in deep learning, featuring a *diagonal* state matrix $\boldsymbol{A}$. To introduce non-linearity and cross-channel mixing, S4 integrates a position-wise linear layer, an activation func-

tion, and a residual connection from input to output. Let $\odot$ represent the element-wise product, and $\mathrm{S4}(\cdot)$ denote the S4 mechanism, where each channel's output follows (3) with its convolutional kernel $\overline{\boldsymbol{K}}^{(d)}$. To facilitate theoretical analysis, certain subtle details—such as activation functions—may differ slightly from those in previous studies (Gu et al., 2022b;a). We define the *deep S4 layer* as:

$$
\boldsymbol{y}_t = \mathrm{ReLU}(\boldsymbol{W} \cdot \mathrm{S4}_t(\boldsymbol{x}_1, \ldots, \boldsymbol{x}_t) + \boldsymbol{\beta} + \boldsymbol{u} \odot \boldsymbol{x}_t), \quad (4)
$$

where $\boldsymbol{W} \in \mathbb{R}^{D \times D}$ and $\boldsymbol{\beta} \in \mathbb{R}^D$ represent the linear projection matrix and bias, respectively, and $\boldsymbol{u} \in \mathbb{R}^D$ is the coefficient of the residual connection. Trainable parameters include SSM parameters $(\boldsymbol{A}^{(d)}, \boldsymbol{B}^{(d)}, \boldsymbol{C}^{(d)}, \Delta^{(d)})$ across $D$ channels with $\boldsymbol{A}^{(d)}$ being diagonal, as well as linear layer $(\boldsymbol{W}, \boldsymbol{\beta})$ and residual connection $\boldsymbol{u}$.

**Selective State Space Models (S6).** All SSMs mentioned above exhibit linear time invariance (LTI), meaning their dynamics remain constant over time. A key limitation of LTI SSMs is their fixed dynamics, hindering selective context extraction and input-dependent state transitions. S6 (Gu & Dao, 2024) addresses this by making parameters input-dependent. At each time step $t$, given input $\boldsymbol{x}_t \in \mathbb{R}^D$, S6 introduces input-dependent step sizes $\boldsymbol{\Delta}_t = (\Delta_t^{(1)}, \ldots, \Delta_t^{(D)})^\top \in \mathbb{R}^D$, input transition vectors $\boldsymbol{B}_t \in \mathbb{R}^{H \times 1}$ and output mapping vectors $\boldsymbol{C}_t \in \mathbb{R}^{1 \times H}$ via linear projection:

$$
\boldsymbol{\Delta}_t = \mathrm{softplus}(\boldsymbol{W}_{\boldsymbol{\Delta}}\boldsymbol{x}_t + \boldsymbol{\beta}_{\boldsymbol{\Delta}}), \quad \boldsymbol{B}_t = \boldsymbol{W}_{\boldsymbol{B}}\boldsymbol{x}_t, \quad \boldsymbol{C}_t = \boldsymbol{W}_{\boldsymbol{C}}\boldsymbol{x}_t,
$$

where the diagonal state matrices $\boldsymbol{A}^{(1)}, \ldots, \boldsymbol{A}^{(D)}$ remain input-independent. The weight $\boldsymbol{W}_{\boldsymbol{\Delta}} \in \mathbb{R}^{D \times D}$ is factorized as $\boldsymbol{W}_{\boldsymbol{\Delta}} = \boldsymbol{W}_{\boldsymbol{\Delta},\downarrow}\boldsymbol{W}_{\boldsymbol{\Delta},\uparrow}$, with $\boldsymbol{W}_{\boldsymbol{\Delta},\downarrow} \in \mathbb{R}^{D \times R}$, $\boldsymbol{W}_{\boldsymbol{\Delta},\uparrow} \in \mathbb{R}^{R \times D}$ to reduce computation (Wang et al., 2021; 2023a). Trainable parameters in S6 include $\boldsymbol{A}^{(d)}$ across $D$ channels, $\boldsymbol{W}_{\boldsymbol{\Delta},\uparrow}, \boldsymbol{W}_{\boldsymbol{\Delta},\downarrow}$ and $\boldsymbol{\beta}_{\boldsymbol{\Delta}}$ for computing $\boldsymbol{\Delta}_t$, and $\boldsymbol{W}_{\boldsymbol{B}}, \boldsymbol{W}_{\boldsymbol{C}} \in \mathbb{R}^{H \times D}$ for computing $\boldsymbol{B}_t, \boldsymbol{C}_t$. Discretization follows: $\overline{\boldsymbol{A}}_t^{(d)} = \exp(\Delta_t^{(d)} \boldsymbol{A}^{(d)}), \overline{\boldsymbol{B}}_t^{(d)} = \Delta_t^{(d)} \boldsymbol{B}_t$. Unlike S4, where $\boldsymbol{B}^{(d)}$ varies per channel, S6's variation on $\overline{\boldsymbol{B}}^{(d)}$ stems from the scalar $\Delta_t^{(d)}$. Additionally, S6 shares $\boldsymbol{C}_t$ for all channels at each time step $t$, while S4 assigns a distinct $\boldsymbol{C}^{(d)}$ to each channel.

**Mamba & Jamba.** Similar to the Transformer block, which consists of attention and linear layers, the Mamba-I block proposed by Gu & Dao (2024) features an S6 module, a point-wise 1D causal convolution layer (Conv1d) for token mixing, linear layers — including input ($\boldsymbol{W}_{\mathrm{in}}$) and output ($\boldsymbol{W}_{\mathrm{out}}$) projection layers and a gated MLP. Mamba-II (Dao & Gu, 2024) further simplifies the state matrix $\boldsymbol{A}$ to be a scalar. Building on Mamba-I, Jamba (Lieber et al., 2025) introduces a hybrid architecture that integrates both Transformer blocks and Mamba blocks, leveraging the strengths of both to enhance performance. This paper

---

[1]Note that $\boldsymbol{B}, \boldsymbol{C}$ are vectors; we use bold capitals for consistency with prior work (Gu et al., 2022b; Gu & Dao, 2024).

focuses on Mamba-I (referred as Mamba in this paper) and Jamba, deferring Mamba-II discussions to the appendix.

## 3.2. Parameter-Efficient Fine-Tuning

**Input-Injection Methods.** Input-injection methods, such as prompt tuning (Lester et al., 2021) and prefix-tuning (Li & Liang, 2021), enhance the model's input by injecting specialized sequences. Prompt tuning prepends a set of trainable embeddings $\boldsymbol{P} \in \mathbb{R}^{D \times M}$ to the original input $\boldsymbol{X} \in \mathbb{R}^{D \times N}$, forming the concatenated sequence $\widetilde{\boldsymbol{X}} = [\boldsymbol{P}; \boldsymbol{X}]$. Prefix-tuning (Li & Liang, 2021) instead injects learnable vectors into the key and value matrices of each attention layer. For a Transformer layer, it prepends prefix states $\boldsymbol{P}^K, \boldsymbol{P}^V \in \mathbb{R}^{L \times D}$ to the original projections:

$$\widetilde{\boldsymbol{K}} = [\boldsymbol{P}^K; \boldsymbol{K}], \quad \widetilde{\boldsymbol{V}} = [\boldsymbol{P}^V; \boldsymbol{V}],$$

where $\boldsymbol{K}$ and $\boldsymbol{V}$ are the key and value matrices derived from the input. We note that prefix-tuning is functionally equivalent to prepending soft tokens to the input at each attention layer and discarding the corresponding outputs associated with the prepended tokens. This view simplifies adaptation to SSMs, which lack explicit key and query projections. Yoshimura et al. (2025) also adopt this implementation, though they refer to it as affix-tuning.

**Architecture-Enhancement Methods.** These methods modify the model's internal structure to introduce tunable components. In the context of SSMs, one example is Additional-scan (Yoshimura et al., 2025), which expands the state dimensions within the SSM block and fine-tunes only the added parameters, leaving the original weights untouched.

**Weight-Tuning Methods.** Notable weight-tuning methods include LoRA (Hu et al., 2021) and its variants (Liu et al., 2024; Hayou et al., 2024), as well as BitFit (Zaken et al., 2022). LoRA (Hu et al., 2021) fine-tunes a model by introducing low-rank updates to its weight matrices. Given a weight matrix $\boldsymbol{W}_0 \in \mathbb{R}^{D \times D}$, LoRA updates it as follows:

$$\boldsymbol{W} = \boldsymbol{W}_0 + \boldsymbol{W}_\downarrow \boldsymbol{W}_\uparrow,$$

with $\boldsymbol{W}_\downarrow \in \mathbb{R}^{D \times R}$, $\boldsymbol{W}_\uparrow \in \mathbb{R}^{R \times D}$, and $R \ll D$ being the rank. Only $\boldsymbol{W}_\downarrow$ and $\boldsymbol{W}_\uparrow$ are trained, reducing the number of trainable parameters from $D^2$ to $2RD$. Weight-*D*ecomposed *L*ow-*R*ank *A*daptation (DoRA) (Liu et al., 2024) improves upon LoRA by decomposing the weight matrix into two components: magnitude ($\boldsymbol{m} \in \mathbb{R}^D$) and direction ($\boldsymbol{W}_\downarrow \boldsymbol{W}_\uparrow$), leading to the formulation

$$\boldsymbol{W} = \boldsymbol{m} \frac{\boldsymbol{W}_0 + \boldsymbol{W}_\downarrow \boldsymbol{W}_\uparrow}{\|\boldsymbol{W}_0 + \boldsymbol{W}_\downarrow \boldsymbol{W}_\uparrow\|}.$$

This additional parameter $\boldsymbol{m}$ enhances both training capacity and stability. LoRA+ (Hayou et al., 2024) modifies

LoRA by applying different learning rates to $\boldsymbol{W}_\downarrow$ and $\boldsymbol{W}_\uparrow$, enabling more effective feature learning. In contrast, Bit-Fit (Zaken et al., 2022) updates only the bias terms, offering a lightweight and highly parameter-efficient alternative.

# 4. Benchmarking PEFT Methods on SSM-based Models

In this section, we examine the effectiveness of popular PEFT methods when applied naively to SSM-based models, specifically Mamba and Jamba.

## 4.1. Experiment Setup

We evaluate PEFT methods across three categories: input-injection, architecture-enhancement, and weight-tuning. For input-injection methods, we use prompt tuning (Lester et al., 2021) and prefix-tuning (Li & Liang, 2021), where prefix-tuning employs an overparameterized MLP for stable optimization. For architecture-enhancement methods, we include Additional-scan (Yoshimura et al., 2025), which introduces and fine-tunes newly added state dimensions in SSM modules. For weight-tuning, we consider BitFit (Zaken et al., 2022) and LoRA★, including LoRA (Hu et al., 2021) and DoRA (Liu et al., 2024), while LoRA+ (Hayou et al., 2024) is deferred to Sec. E.2. BitFit fine-tunes the bias terms of Conv1d and $\boldsymbol{W}_{\boldsymbol{\Delta},\uparrow}$.

We use six datasets spanning different domains: GLUE for natural language understanding (Wang et al., 2019), DART for RDF-to-text generation (Nan et al., 2021), SAM-Sum (Gliwa et al., 2019) for summarization, Spider for text-to-SQL generation (Yu et al., 2018), and two vision datasets—CIFAR-10 (Krizhevsky et al., 2009) and CelebA (Liu et al., 2015), with the vision datasets processed by cropping, resizing, and flattening pixel values into space-separated numerical sentences. Details are in Sec. B. Prefix-tuning requires significantly more parameters than other PEFT methods due to its per-layer MLP for projecting fixed sequences into soft tokens. For all methods—except prefix-tuning and the special case of LoRA and DoRA when applied to both linear projection layers—we limit trainable parameters to below 1% for Mamba and below 0.15% for Jamba. For Jamba, all PEFT methods are applied to Mamba layers, while Transformer layers remain frozen to isolate performance effects. See more details in Sec. C.1.

## 4.2. Results

Table 1 summarizes the benchmarking results. Detailed results for GLUE and Spider subtasks appear in Sec. C.2. We analyze the results from three key perspectives below.

| Model | Method | Major Target Module | GLUE | DART | | SAMSum | | | Spider | CIFAR-10 | CelebA |
|-------|--------|---------------------|------|------|------|--------|------|------|--------|----------|--------|
| | | | Avg. Score | METEOR | BLEU | R1 | R2 | RL | Acc. | Acc. | Acc. |
| Mamba | Prompt Tuning | Other | 63.8 | 66.2 | 39.8 | 50.1 | 25.6 | 41.6 | 43.6 | 30.4 | 82.5 |
| | Prefix-Tuning | SSM | 68.6 | 66.6 | 42.5 | 50.6 | 26.5 | 42.1 | 39.7 | 41.0 | 86.5 |
| | BitFit | Both | 76.8 | 67.0 | 43.7 | 50.3 | 25.7 | 41.9 | 48.4 | 44.4 | 86.9 |
| | LoRA | SSM | 76.9 | 68.8 | 48.0 | 50.4 | 26.0 | 41.8 | 55.0 | 52.3 | 87.0 |
| | | LinProj | **81.2** | **70.9** | 49.5 | 50.9 | _27.0_ | 42.3 | 57.5 | **61.0** | 87.0 |
| | | Both | 80.3 | 70.2 | **52.2** | 50.7 | 26.8 | 42.4 | 57.0 | _58.4_ | **89.8** |
| | DoRA | SSM | 77.9 | 68.3 | 47.3 | 48.1 | 24.2 | 39.6 | 55.3 | 44.5 | 87.1 |
| | | LinProj | _81.1_ | 70.7 | **51.6** | _51.0_ | 26.9 | _42.8_ | 60.7 | 57.6 | 86.7 |
| | | Both | 80.8 | _70.8_ | 51.4 | **51.3** | **27.2** | **43.0** | _58.1_ | 58.2 | **89.8** |
| | Additional-Scan | SSM | 62.4 | 60.6 | 15.8 | 37.6 | 17.5 | 30.9 | 26.9 | 32.2 | 86.0 |
| | Full Fine-Tuning | Both | 80.5 | 71.0 | 51.8 | 51.2 | 27.3 | 42.9 | 66.2 | 60.0 | 89.4 |
| Jamba | Prompt Tuning | Other | 73.3 | 54.1 | 6.3 | _54.7_ | 31.8 | 46.8 | **74.9** | 40.9 | 85.6 |
| | Prefix-Tuning | SSM | 56.9 | 59.6 | 14.4 | 11.5 | 1.8 | 10.4 | 0.3 | 29.9 | 82.2 |
| | BitFit | Other | **75.2** | 59.2 | 14.8 | _54.7_ | 31.9 | _47.0_ | _73.7_ | 45.6 | 86.3 |
| | LoRA | LinProj | _73.9_ | **68.9** | **37.8** | 54.6 | **32.3** | 46.8 | 69.3 | **59.7** | **89.0** |
| | DoRA | LinProj | 71.4 | _68.1_ | _28.8_ | **55.2** | _32.2_ | **47.3** | 70.9 | _58.6_ | **89.0** |
| | Additional-Scan | SSM | 68.3 | 63.3 | 20.1 | 53.4 | 30.5 | 45.6 | 69.3 | 50.6 | 0.0 |

*Table 1.* **Benchmarking popular Parameter-Efficient Fine-Tuning (PEFT) methods on Mamba (Gu & Dao, 2024) and Jamba (Lieber et al., 2025) across six real-world datasets.** R1/R2/RL stand for ROUGE-1/2/L. We evaluate PEFT applied to different target modules: SSM module only, linear projection matrices (LinProj) only, both, or other components such as embedding layer. For both Mamba and Jamba, all methods use fewer than 1% and 0.15% of parameters, respectively, except when the target module for LoRA or DoRA is set to "Both" or when prefix-tuning is applied. Comprehensive hyperparameter tuning was performed for all methods. Bold values indicate the best performance for each model (Mamba and Jamba) separately, while underlined values denote the second-best performance for each task, excluding full fine-tuning. Key findings include: (i) among PEFT methods applied to SSM modules, LoRA$^\star$ outperforms others, (ii) for all PEFT methods, LoRA$^\star$ achieves the best performance, (iii) applying LoRA$^\star$ to linear projections yields results comparable to applying it to both linear projections and SSM modules, while outperforming its application solely to SSM modules, and (iv) input-injection methods (i.e., prompt tuning and prefix-tuning), are generally ineffective.

**Superiority of LoRA$^\star$.** The most prominent finding is that LoRA$^\star$ consistently outperforms other PEFT methods (e.g., prompt tuning, prefix-tuning, BitFit, additional-scan), regardless of the target module.

> **Finding:** *Across all target modules, LoRA$^\star$ surpasses existing PEFT methods in performance.*

Even when restricted to SSM modules, LoRA$^\star$ still outperforms all other PEFT baselines applied to the same target.

**Limitations of Input-Injection Methods.** Input-injection methods like prefix-tuning are ineffective for SSM-based models (Table 1), as their expressiveness reduces to tuning only the initial hidden state (Proposition 1). Formal statement, proof and empirical verification are in Sec. C.3.

**Optimal Application of LoRA$^\star$ in SSM-based Models.** Table 1 shows that LoRA$^\star$ outperforms all other PEFT methods in most scenarios. From our results, we explore the optimal layers for applying LoRA$^\star$ in SSM-based models: the SSM module, the linear projection matrices, or a combi-

nation of both. Note that S6 in Mamba and Jamba includes fine-grained parameters like x_proj ($W_B, W_C, W_{\Delta,\downarrow}$) and dt_proj ($W_{\Delta,\uparrow}$), which were already explored by Yoshimura et al. (2025) on Mamba. We defer a deeper discussion of them to Sec. C.4 and focus on the key question here: Is applying LoRA$^\star$ to SSM modules necessary for performance gains? By narrowing our scope, we aim to clarify LoRA$^\star$'s impact across major components (e.g., SSM modules, linear projection matrices) rather than all specific parameters.

We evaluate LoRA$^\star$'s performance on linear projections using $W_{in}$, $W_{out}$, and both combined. Since the performance of different combinations of linear projections is consistent across datasets (see Sec. C.4.), we only report the results for LoRA$^\star$ applied to $W_{in}$ in Table 1. For SSM modules, we apply LoRA$^\star$ to weight matrices, including those for the input-dependent step size $\Delta$. For state transition matrices $A$, we treat their diagonal structures as vectors, concatenate them across channels to form a matrix, and apply LoRA$^\star$. Table 1 summarizes results for the best-performing configurations (see Sec. C.2 for full results). Based on these results, we present the following finding:

> **Finding:** *For LoRA⋆: Tuning on SSMs is less effective than tuning linear projection matrices, with the latter performing comparably to tuning both.*

Detailed experiments, including LoRA⋆ on different linear projection matrices and additional evaluations of Mamba-II, are presented in Sec. C.2. These experiments reinforce the finding that LoRA⋆ is highly effective for linear projections but less suitable for SSM modules.

To further elucidate this concept, we present the following lemma, which examines a simplified model architecture consisting of S6 with two linear input projection matrices at each layer. We demonstrate that fine-tuning one input projection matrix encompasses the expressivity of fine-tuning the parameters $W_B$, $W_C$, and $W_{\Delta,\uparrow}$. Consider an S6 model with two input projection matrices $W_{\text{in},1}, W_{\text{in},2} \in \mathbb{R}^{D \times D}$: the first affects how internal parameters depend on the input, while the second governs the input passed directly into the S6 module. Under this setup, the output $y_N^{(d)}$ can be expressed as:

$$y_N^{(d)} = \underbrace{\overbrace{C(W_{\text{in},1}x_N)^\top}^{\text{Input-dependent }C_N} \sum_{n=1}^{N} \left( \prod_{m=1}^{n} \overbrace{\overline{A}(W_{\text{in},1}x_m)}^{\text{Input-dependent }\overline{A}_m} \right) \overbrace{\overline{B}_n(W_{\text{in},1}x_n)}^{\text{Input-dependent }\overline{B}_n}}_{\text{Parameters depending on input after projection }W_{\text{in},1}} \underbrace{(W_{\text{in},2}x_n)^{(d)}}_{\text{Input after projection }W_{\text{in},2}}.$$

When $W_{\text{in},1} = W_{\text{in},2}$, this reduces to a standard architecture with a single input projection followed by an S6 layer. For simplicity, we let $\beta_\Delta = 0$. Then the full model is parameterized by $(\{A^{(d)}\}_{d=1}^{D}, W_B, W_C, W_{\Delta,\uparrow}, W_{\Delta,\downarrow}, W_{\text{in},1}, W_{\text{in},2})$. Assume none of the parameters are zero and $D > 2H + R$, where $R$ is the rank of $W_{\Delta,\downarrow}W_{\Delta,\uparrow}$.

**Lemma 1** (Expressivity of Fine-Tuning Projection Matrices). *Consider two models with the architecture described above. Let:*

- *A target model $f^\star$ parameterized by $(\{A^{\star(d)}\}_{d=1}^{D}, W_B^\star, W_C^\star, W_{\Delta,\uparrow}^\star, W_{\Delta,\downarrow}^\star, W_{in,1}^\star, W_{in,2}^\star)$;*

- *A frozen model $f_0$ parameterized by $(\{A^{\star(d)}\}_{d=1}^{D}, W_B, W_C, W_{\Delta,\uparrow}, W_{\Delta,\downarrow}^\star, W_{in,1}, W_{in,2}^\star)$.*

*The two models share $\{A^{\star(d)}\}_{d=1}^{D}$, $W_{\Delta,\downarrow}^\star$, and $W_{in,2}^\star$, while differing in $W_B$, $W_C$, $W_{\Delta,\uparrow}$, and $W_{in,1}$. Then, there exists an updated projection matrix $\widehat{W_{in,1}}$ such that the frozen model matches the output of the target model without updating $W_B$, $W_C$, $W_{\Delta,\uparrow}$ for any input sequence, i.e.,*

$$f(\cdot; \{A^{\star(d)}\}_{d=1}^{D}, W_B, W_C, W_{\Delta,\uparrow}, W_{\Delta,\downarrow}^\star, \widehat{W_{in,1}}, W_{in,2}^\star)$$
$$= f^\star(\cdot; \{A^{\star(d)}\}_{d=1}^{D}, W_B^\star, W_C^\star, W_{\Delta,\uparrow}^\star, W_{\Delta,\downarrow}^\star, W_{in,1}^\star, W_{in,2}^\star).$$

We expand on this discussion in Sec. C.4, where we present both theoretical proofs and empirical validation. The lemma shows that tuning the linear projection matrix can match

the expressive power of certain SSM parameters (i.e., $W_B$, $W_C$, and $W_{\Delta,\uparrow}$), aligning with our empirical observation that tuning only the linear projections already performs well. However, a key limitation of tuning only the linear projection matrices remains: such tuning lacks the expressive power to affect the state matrix $A$, which is an essential parameter for sequence-to-sequence operations. Therefore, tuning the SSM modules is still necessary. Existing PEFT methods fall short in effectively tuning SSM modules: (i) alternative methods underperform compared to LoRA⋆ on SSM modules, and (ii) applying LoRA⋆ to SSM modules does not improve performance beyond applying it to linear projections alone. These findings highlight a gap in current PEFT techniques for SSM modules, leading to an importantca question: *Is there a more effective strategy for fine-tuning SSM modules?*

## 5. Sparse Dimension Tuning

This section aims to develop an algorithm for tuning SSM modules. In doing so, we start by first analyzing the roles of different parameters, as outlined in Lemma 2. This analysis motivates us to classify channels and state dimensions into three categories: (i) zero, (ii) trainable, and (iii) frozen, leading to the development of the *Sparse Dimension Tuning and Pruning* (SDT-P) method. We then establish theoretical guarantees for applying SDT-P to SSM modules and LoRA to linear projection matrices (Theorem 1). Finally, we simplify SDT-P into *Sparse Dimension Tuning* (SDT) by omitting pruning, as pruned parameters can be effectively considered as being trained to zero. This simplified version serves as the primary method used in our experiments.

### 5.1. Understanding Key Parameters in S4 Modules

**Problem Setting.** Inspired by the work by Zeng & Lee (2024), we analyze the expressive power of S4 parameters using a similar framework. We assume a well-performing target model and a frozen model (pretrained or random) and aim to update the frozen model efficiently to match the target. Following Zeng & Lee (2024), we assume the frozen model has a capacity at least as large as the target model. This assumption ensures analytical tractability and is reasonable, as frozen models are typically overparameterized in practice. Both models are S4 with hidden dimensions $H_\star$ (target) and $H \geq H_\star$ (frozen). Assuming all hidden dimensions are active (i.e., all parameters are non-zero), we define their dynamics using discretized parameters $(\overline{A}, \overline{B}, C)$:

$$\text{(Target model)} \quad f^\star(x)_n = \sum_{m=1}^{n} C_\star \overline{A}_\star^{m-n} \overline{B}_\star x_m,$$
$$\text{(Frozen model)} \quad f_0(x)_n = \sum_{m=1}^{n} C_0 \overline{A}_0^{m-n} \overline{B}_0 x_m,$$

where $\text{diag}(\overline{A}_\star), \overline{B}_\star, C_\star \in \mathbb{R}^{H_\star}$, $\text{diag}(\overline{A}_0), \overline{B}_0, C_0 \in \mathbb{R}^{H}$. This formulation shows that the S4 module remains unchanged even if the state dimensions are permuted.

**Parameter Efficiency Analysis on S4.** We analyze the parameter efficiency of updating a frozen S4 module after discretizing its parameters $(\overline{A}_0, \overline{B}_0, C_0)$ to match the functionality of a target S4 module with discretized parameters $(\overline{A}_\star, \overline{B}_\star, C_\star)$. Based on this setup, we present the following result characterizing the minimum number of parameters that must be tuned for functional equivalence.

**Lemma 2** (Minimal Parameter Adjustment for S4 Fine-Tuning). *Assume all hidden dimensions of the target model $f^\star$ are non-zero, i.e., all elements of $\mathrm{diag}(\overline{A}_\star) \odot \overline{B}_\star \odot C_\star$ are non-zero. To update frozen model $f_0$ such that it becomes functionally equivalent to the target model $f^\star$, the minimum number of tunable parameters is:*

$$
\min_{\overline{A}, \overline{B}, C} \overbrace{\left\| \left[ \mathrm{diag}(\overline{A}) \odot \overline{B} \odot C^\top \right]_{(H_\star+1):H} \right\|_0}^{\text{eliminating redundant dimensions}}
$$
$$
+ \underbrace{\left\| \left[ \overline{A} \right]_{1:H_\star, 1:H_\star} - \overline{A}_\star \right\|_0 + \left\| \left[ \overline{B} \odot C^\top \right]_{1:H_\star} - \overline{B}_\star \odot C_\star^\top \right\|_0}_{\text{aligning remaining dimensions with target model}}, \tag{5}
$$

*subject to*

$$(\overline{A}, \overline{B}, C) \in \{ (P^\top \overline{A}_0 P, P^\top \overline{B}_0, C_0 P) : P \text{ is a permutation matrix} \}.$$

Note that the search space consists of all possible S4 parameterizations that can be obtained by permuting the hidden dimensions of the frozen model. Proofs and further details are provided in Sec. D.1. This result highlights three distinct roles of the state dimensions. First, any dimensions that do not contribute to the target function (represented by the first term in (5)) are effectively zero and can be pruned. These correspond to state dimensions larger than those of the target model after permutation, indicating that redundant information can be directly removed to eliminate its impact. Second, among the remaining dimensions, alignment is necessary for those that do not already match the target. The state matrix $A$ plays a crucial role in sequence modeling by capturing dependencies between tokens at different positions. To achieve functional equivalence (as represented by the second term in (5)), $A$ must be aligned. Notably, dimensions that are already aligned with the target require no updates. These two insights motivate our Sparse Dimension Tuning and Pruning (SDT-P) method, which classifies hidden dimensions into three categories: (i) zero, (ii) frozen (already aligned), and (iii) trainable. Finally, the third term in (5) indicates that the expressive power of $\overline{B}$ and $C$ is essentially equivalent, meaning that tuning either one is equivalent to updating both.

### 5.2. Sparse Dimension Tuning and Pruning (SDT-P)

Building on Lemma 2, we introduce SDT-P, the precursor to Sparse Dimension Tuning (SDT). SDT-P updates parameters selectively based on the role of each state dimension. In the multi-channel case, we first categorize the channel

dimensions into three groups: pruned, frozen, and trainable. Then, the state dimensions of each trainable channel are also categorized as pruned, frozen, or trainable. This hierarchical selection ensures that updates are applied only when necessary, while pruned dimensions are discarded and frozen dimensions remain unchanged.

**Dimension Selection Algorithm.** To enable this structured tuning process, we first introduce our dimension selection algorithm. The algorithm starts with a warmup epoch, where the SSM modules are updated using a subset of the dataset for one epoch. After this warmup, we classify channel dimensions based on the magnitude of the state matrix $A$: dimensions with small magnitude are pruned (set to zero), those with significant changes are marked as trainable, and the rest remain frozen. Next, we apply the same classification to state dimensions, but only within the trainable channels. The detailed pseudo-code is in Sec. D.4.

**Parameter Update Scheme.** Once the channel and state dimensions are selected, we determine how to update the parameters. **(S4)** For S4, Gu et al. (2022a) showed that tuning $C$ alone is as effective as tuning both $\overline{B}$ and $C$. Therefore, we always freeze $\overline{B}$ and update only $\overline{A}$ and $C$. Specifically, an entry in $\overline{A}$ or $C$ is trainable if and only if both its channel and state dimensions are trainable. If either the channel or state dimension is pruned, the entry is pruned as well. All other entries remain frozen. **(S6)** For S6, where parameters are input-dependent, we update $\overline{A}, W_B$, and $W_C$ instead. Since $W_B$ and $W_C$ operate across channels, we categorize their updates based only on channel dimensions—we do not update individual state dimensions differently for each channel. Based on this categorization, we mark the corresponding columns of $W_B$ and $W_C$ as trainable, frozen, or pruned accordingly.

The dimension selection algorithm and parameter updates together form the SDT-P method for tuning SSM modules. Next, we provide theoretical guarantees for applying SDT-P to SSM modules and LoRA$^\star$ to linear projection matrices.

### 5.3. Expressive Power of SDT-P Combined with LoRA

Our analysis focuses on simplified SSM-based models, where each layer consists of an SSM module followed by linear projection matrices with residual connections. We refer to this structure as a deep SSM layer: i) a deep S4 layer consists of an S4 module followed by linear projections; ii) a deep S6 layer follows the same structure but replace S4 with S6. A deep S4 model is composed of deep S4 layers, while a deep S6 model consists of deep S6 layers. The detailed formulation of deep S4 layers is provided in Sec. 3, and a deep S6 layer follows the same structure with S4 replaced by S6. The following theorem highlights the expressive power of SDT-P on updating SSM modules, where each layer uses a single type of SSM module (S4 or S6) followed by linear projections.

**Theorem 1** (Expressive Power of SDT-P with LoRA on Simplified SSM-based Models). *Assume all layers use linear activations. Let $f_0$ be a frozen deep S4 or S6 model with $L$ layers, each containing $H$ hidden states per channel. Let $f^\star$ be a smaller target model of the same type (S4 or S6), with no residual connections, $L^\star < L$ layers, and $H^\star < H$ hidden states per channel. Then, there exists a set of parameter updates to $f_0$ satisfying the following conditions such that for any finite-length input sequence $\boldsymbol{X} = (\boldsymbol{x}_1, \ldots, \boldsymbol{x}_N)$ with $\boldsymbol{x}_n \in \mathcal{X} \subset \mathbb{R}^D$, where $\mathcal{X}$ is bounded, the resulting model $f$ satisfies $f(\boldsymbol{X}) = f^\star(\boldsymbol{X})$:*

1. *(**SDT-P on SSM**) In each SSM module, update at most $\lceil DL^\star/L \rceil$ channels. Within each updated channel, fine-tune at most $H^\star$ hidden states and set the rest to zero.*
2. *(**LoRA$^\star$ on Linear Projections**) Apply rank-$\lceil L/L^\star \rceil$ updates to each linear projection matrix.*
3. *(**Minimal Additional Updates**) Update only the residual connections, per-layer biases, and the final-layer output projection matrix.*

For proof and details, refer to Sec. D.2 and D.3. This theorem shows that a larger pretrained model can be fine-tuned into any smaller model of the same architecture by applying SDT-P to SSM modules and LoRA$^\star$ to linear projection matrices. Moreover, for less complex tasks, where the target model has fewer layers ($L^\star$) and hidden states ($H^\star$), the required number of trainable channels and hidden states also decreases. This aligns with the theoretical analysis of LoRA by Zeng & Lee (2024), which demonstrates that larger pretrained models require fewer learnable parameters (i.e., a lower-rank update) during fine-tuning, especially for simpler tasks. While our theorem assumes linear activations, no residual connections in the target model, and full fine-tuning of the last-layer projection matrix, our findings have broader implications. As our experimental results in Sec. 6 will show, these insights generalize beyond these theoretical constraints.

---

**Algorithm 1** Dimension Selection Algorithm of SDT
---
**Input:** A small subset of dataset $\mathcal{D}$, warmup epochs $E$, number of layers $L$, total channels $D$, total states $H$, channel freeze ratio $\alpha$, state freeze ratio $\beta$

```
/* Warmup epochs                              */
```
Perform full update on SSM modules using $\mathcal{D}$ for $E$ epochs
  **for** $l = 1$ **to** $L$ **do**
```
    /* Unfreeze dimensions                     */
```
    Sort channels $\mathbb{D}$ based on changes of $\|\overline{\boldsymbol{A}}^{(d)}\|$
    Freeze the bottom $\beta|\mathbb{D}|$ channels, denoted by $\mathbb{D}'$
    **for** $d \in \mathbb{D}'$ **do**
      Sort state dimensions by the changes in $\|\overline{\boldsymbol{A}}^{(d)}\|$
      Freeze the bottom $\alpha|\mathbb{H}|$ state dimensions at the $d$-th channel

---

## 5.4. Sparse Dimension Tuning (SDT): A Pruning-Free Alternative

While SDT-P classifies channels and states into three categories, we simplify our approach by omitting pruning and categorizing parameters as either trainable or frozen. We refer to this simplified method as *Sparse Dimension Tuning* (SDT). This reduces the number of hyperparameters, as pruned parameters are effectively equivalent to being trained to zero. The resulting dimension selection approach is outlined in the pseudo-code (Alg. 1), which corresponds to the update scheme illustrated in Fig. 1. Experiments will show that this simplification remains effective.

**Overhead Analysis.** We assess the computational overhead of applying SDT with LoRA (for linear projection matrices) versus LoRA alone with Table 2 summarizing the results. Although SDT involves an additional dimension selection stage, Table 2 shows that this incurs minimal extra cost. Furthermore, with the same parameter budget, SDT for SSM modules combined with LoRA on linear projections runs faster than LoRA alone, since LoRA introduces extra matrix multiplications between two low-rank matrices for the SSM modules, whereas SDT does not. In Sec. D.6, we detail the experimental settings and present a memory usage analysis showing that SDT also consumes less memory during fine-tuning for the same reason.

| Stage | Method | Mamba-130M | Mamba-1.4B | Jamba-Mini-52B |
|---|---|---|---|---|
| Dim. Selection | LoRA & SDT | $16.5 \pm 3.9$ | $85.8 \pm 5.3$ | $163.9 \pm 10.2$ |
| Training | LoRA | $410.0 \pm 80.0$ | $2060.0 \pm 135.0$ | $3427.5 \pm 185.0$ |
| (per epoch) | LoRA & SDT | $330.0 \pm 77.5$ | $1697.5 \pm 87.5$ | $3065.0 \pm 232.5$ |

*Table 2.* **PEFT combining SDT with LoRA is more efficient than LoRA alone when the same number of trainable parameters are used.** Shown are dimension selection and per-epoch training times (s) for Mamba and Jamba models.

## 6. Experimental Studies of SDT

In this section, we evaluate the performance of SDT in tuning SSM modules, comparing it to LoRA$^\star$, the best existing PEFT method for fine-tuning SSM modules, as shown in Sec. 4. Our experiments reveal the key result:

> **Finding:** *SDT outperforms LoRA$^\star$ on SSM modules.*

### 6.1. Synthetic Experiments on Deep S4 Models

This experiment validates our theoretical guarantees under broader conditions, including residual connections and ReLU activations in both models, without fully fine-tuning the last-layer projection matrix. See Sec. E.1 for details.

**(Experimental Setup)** We employ a regression setting to validate our theoretical results. We randomly ini-

tialize two models: a one-layer deep S4 model as the target and a four-layer deep S4 model as the frozen model. LoRA is applied to linear projection matrices, while different methods are tested on the SSM module to assess their effectiveness. The goal is to update the frozen model to match the target model's functionality.

We generate an input sequence $X$ of length 200 and dimension 64, with values uniformly drawn from integers between 0 and 9. This input is then processed through the target model to obtain the corresponding outputs. These input-output pairs are used to train the frozen model over 500 iterations using the Mean Squared Error (MSE) loss. **(Results)** Figure 2 shows the MSE, averaged across all tokens, plotted against the number of trainable parameters for different methods on SSM modules. SDT achieves significantly lower MSE than LoRA on SSM modules, demonstrating its effectiveness in updating SSM modules.

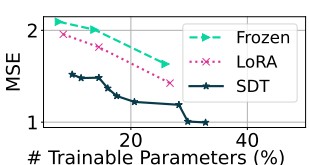

*Figure 2.* SDT surpasses LoRA in tuning S4 within deep S4 models when LoRA is applied to linear projection matrices in synthetic experiments.

### 6.2. Real-World Experiments on Pretrained Models

Lastly, we conduct experiments to evaluate our approach on pretrained models, including Mamba and Jamba with different model sizes. We consider five datasets: GLUE, DART, SAMSum, Spider, and CelebA. For these experiments, we split the datasets into three parts: train, validation, and test, different from benchmarking experiments. We combine our proposed SDT with LoRA$^\star$ and evaluate it in three different settings against three pure LoRA$^\star$ settings. In SDT, 99% of channels are frozen, and we adjust state freeze ratios. For the pure LoRA$^\star$ settings, we apply LoRA$^\star$ to different parameter sets, selecting ranks to ensure all settings have a comparable parameter budget for fair comparison. Residual connections and biases are frozen and learning rates are independently selected via a small grid search over data subsets. See Sec. E.2 for further details.

**Mamba.** The experimental results of Mamba are reported in Table 3, showing that applying SDT on SSM modules outperforms pure LoRA$^\star$, even when 99% of the channels are frozen. This underscores the effectiveness of SDT on fine-tuning SSM modules.

**Jamba.** We extend our experiments to Jamba, applying all tested methods exclusively to its Mamba layers. Notably, the performance gain on Jamba is smaller compared to Mamba. This is because we freeze all Transformer layers to isolate the effect of Mamba layers for a fair evaluation. Additionally, since the Mamba layers in Jamba contain sig-

| LinProj | S6 | GLUE Avg. | DART BLEU | DART MET. | CelebA Acc. | SAMSum R1 | SAMSum R2 | SAMSum RL | Spider Acc. |
|---|---|---|---|---|---|---|---|---|---|
| LoRA | LoRA | 80.8 | 51.0 | 70.2 | **88.6** | 51.6 | **28.2** | 43.2 | 83.5 |
| | SDT | **81.1** | **51.5** | 70.5 | **88.6** | **51.7** | 28.1 | **43.4** | **84.5** |
| DoRA | DoRA | 80.1 | 51.2 | 70.4 | 88.4 | 51.8 | 28.0 | 43.4 | 83.8 |
| | SDT | 78.2 | **51.5** | **70.8** | **88.6** | **52.1** | **28.3** | **43.7** | **85.1** |

*Table 3.* **Performance comparison between SDT and LoRA on pretrained Mamba models. Bold** numbers indicate the best performance for each task. We use Mamba-130M to compare the performance of SDT and LoRA on GLUE (Wang et al., 2019), DART (Nan et al., 2021), and CelebA (Liu et al., 2015) benchmarks. For all other datasets, we employ Mamba-1.4B. We report only the best setting out of three for each method. We observe that SDT outperforms LoRA$^\star$ on updating SSM modules on Mamba.

nificantly fewer parameters than those in the Mamba model, fine-tuning them yields limited performance improvements. Nevertheless, results on GLUE (Table 4) validate the effectiveness of our method. See Table 22 for more results.

| LinProj | S6 | RTE | MRPC | CoLA | SST-2 | QNLI | QQP | MNLI | Avg. |
|---|---|---|---|---|---|---|---|---|---|
| DoRA | DoRA | 65.7 | **77.8** | 7.1 | 93.9 | 77.8 | 67.8 | 85.4 | 67.9 |
| | SDT | **67.1** | 77.5 | **7.5** | **94.2** | **79.6** | **72.7** | **85.5** | **69.2** |

*Table 4.* **Performance comparison between SDT and DoRA on pretrained Jamba models. Bold** numbers indicate the best performance for each task. We use Jamba-Tiny-319M to compare the performance of SDT and DoRA on the GLUE (Wang et al., 2019) benchmark. We report only the best setting out of three for each method. We observe that SDT outperforms DoRA on updating SSM modules on Jamba.

## 7. Discussion

In this paper, we study PEFT methods applied to SSM-based models. Our evaluation of existing PEFT methods provides valuable insights and guidelines for future researchers to parameter-efficiently fine-tune SSM-based models for other domains. Moreover, we take an initial step in establishing a theoretical framework for studying PEFT methods on SSM-based models. Additionally, we introduce SDT, a PEFT method specifically tailored to SSM modules, demonstrating superior performance compared to existing approaches.

**Limitations & Future Works.** The theoretical guarantees for SDT are restricted to linear activations and require full fine-tuning of the last layer. Nonetheless, our experiments show that SDT performs well in practice despite these constraints. Addressing these theoretical limitations or developing new PEFT methods applicable to broader scenarios is a promising future direction. Additionally, our theory shows that modifying a subset of channels and states is sufficient but does not guide optimal selection. Our approach, based on a warmup stage and parameter magnitude, might not be optimal. Future research could explore the impact of channel/state selection and improve dimension selection algorithms.

## Impact Statement

This paper presents work whose goal is to advance the field of Machine Learning. There are many potential societal consequences of our work, none which we feel must be specifically highlighted here.

## Acknowledgment

The works is supported by NSF Award DMS-2023239, NSF CAREER Award CCF-2339978, Amazon Research Award, and a grant from FuriosaAI.

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

# Appendix

# A. Additional Related Works

## A.1. Additional Related Works on SSMs

Linear State-Space Layers (LSSL) represent one of the earliest SSM layers utilized in deep learning, functioning as continuous-time, recurrent, and convolutional models (Gu et al., 2021). LSSL employs HiPPO theory (Gu et al., 2020) to initialize the state matrix $A$, enabling the capture of long dependencies. However, LSSL is computationally expensive, limiting its practical application. Gu et al. (2022b) introduced Structured State Space Models (S4), which optimize computation efficiency by employing a structured state matrix $A$. Gupta et al. (2022) proposed DSS, which simplifies the model by using a diagonal matrix for $A$ and empirically demonstrated that it suffices to achieve performance comparable to S4. Further, Gu et al. (2022a) provided a theoretical explanation for the effectiveness of the diagonal state matrix $A$ in DSS and introduced S4D, which offers various initialization methods for $A$. Subsequently, the diagonal structure of the state matrix $A$ has been adopted in follow-up methods (Gu & Dao, 2024). Despite differences in optimization algorithms, we refer to S4 and its close variants, including DSS and S4D, collectively as S4. This terminology encompasses models that maintain the standard discrete-time SSM form with a diagonal state matrix.

Despite of the remarkable performance of SSMs on certain tasks of sequence modeling, SSMs still showed worse performance than Transformers on language modeling. Fu et al. (2022) transitioned from synthetic language modeling tasks to real language modeling tasks with SSMs. They proposed H3, which is inspired by Linear Attention (Katharopoulos et al., 2020), introducing both diagonal SSM and shift SSM. Recently, Mamba (Gu & Dao, 2024; Dao & Gu, 2024) escaped from linear time invariance (LTI) modeling by introducing input-dependent terms and achieved better performance than Transformers on language modeling. Furthermore, several hybrid models (Lieber et al., 2025; Park et al., 2024) tried to exploit the advantages of both SSMs and Transformers.

## A.2. Additional Related Works on PEFT

In this section, we provide a more detailed description of the baseline methods.

**LoRA (Hu et al., 2021).** LoRA (Low-Rank Adaptation) focuses on fine-tuning large models by freezing pretrained parameters and injecting trainable low-rank matrices into each layer of the Transformer architecture. The intuition behind using low-rank matrices comes from linear algebra, where a large matrix can be closely approximated by the product of two smaller matrices. The number of trainable parameters can be controlled with the rank of the low-rank matrices. LoRA also uses a scaling parameter (LoRA alpha) for the weight matrices to control the balance of the original model weights and LoRA weights during training. After fine-tuning, LoRA weights can be merged with the original model weights, introducing no additional inference overhead.

**Prompt Tuning (Lester et al., 2021).** Prompt tuning freezes all model weights and prepends a trainable soft prompt to the input prompt. The soft prompt consists of trainable virtual tokens, which are continuous. At inference time, prompt tuning introduces an inference overhead based on the number of virtual tokens used.

**Prefix-Tuning (Li & Liang, 2021).** Prefix-tuning also prepends trainable tokens to the input like prompt tuning but injects separate prefixes in every layer. For each Transformer layer, prefix-tuning prepends trainable embeddings to the attention's $K$ and $V$ matrix. The authors have found that directly training these prefixes can lead to unstable training, so they propose to over-parameterize them with a large MLP to increase training stability. After training, the MLP can be dropped. Like prompt tuning, prefix-tuning introduces an inference overhead, scaling with the number of trainable embeddings.

**BitFit (Zaken et al., 2022).** BitFit is a simple but effective PEFT method that freezes all model weights except the bias terms, consequently greatly reducing the number of trainable parameters. As no additional parameters are added, no inference overhead occurs.

**Theoretical Understanding of PEFT.** Numerous efforts have been made to theoretically understand existing PEFT methods. For input-injection methods, Wang et al. (2023b), Petrov et al. (2024), and Oymak et al. (2023) have theoretically analyzed the effectiveness and limitations of prompt tuning and prefix-tuning for Transformer-based models. For LoRA, Zeng & Lee (2024) explored its expressive power by demonstrating that even a randomly initialized model can be adapted to match any smaller target model using LoRA. Some of our theoretical analysis draws upon the framework established by

Zeng & Lee (2024). Jang et al. (2024) conducted a theoretical exploration of LoRA within the neural tangent kernel (NTK) regime.

## B. Details of Datasets

In this paper, we consider six datasets across three domains: (i) Natural Language Understanding (NLU), represented by GLUE (Wang et al., 2019); (ii) Natural Language Generation (NLG), including SAMSum (Gliwa et al., 2019), Spider (Yu et al., 2018) and DART (Nan et al., 2021); and (iii) Computer Vision (CV), represented by CIFAR-10 (Krizhevsky et al., 2009) and CelebA (Liu et al., 2015).

**GLUE (Wang et al., 2019).**    The GLUE (General Language Understanding Evaluation) benchmark is a collection of datasets used for training, evaluating, and analyzing natural language understanding models across a range of diverse tasks. The benchmark includes nine sentence- or sentence-pair language understanding tasks that require various features of understanding, such as sentiment analysis, linguistic acceptability, semantic textual similarity, and question answering. We use seven datasets from the GLUE benchmark (RTE, MRPC, CoLA, SST-2, QNLI, QQP, MNLI) where the model has to choose between two or three (for MNLI) different choices for the respective task. Except for CoLA, we evaluate all used datasets with the accuracy metric. For CoLA, Matthews correlation is employed.

**SAMSum (Gliwa et al., 2019).**    SAMSum is a dataset for dialogue summarization research, comprising approximately 16,000 synthetic text conversations with accompanying summaries. Created by English-fluent linguists, these exchanges simulate real-world digital communications across various topics and styles. The conversations range from informal to formal, incorporating elements like slang and emoticons to reflect authentic messaging patterns. Each dialogue is paired with a concise, third-person summary, capturing its essential content. This structure makes SAMSum particularly useful for developing and evaluating automated summarization systems capable of processing conversational text.

**Spider (Yu et al., 2018).**    Spider is a large-scale, complex, and cross-domain semantic parsing and text-to-SQL dataset. It contains about 10,000 annotated SQL queries, distributed across 200+ databases, each with multiple tables. We follow Scholak et al. (2021) and use about 7,000 examples for training and about 1,000 examples for validation, where we ignore sequences longer than 1536 tokens. The dataset consists of English question and SQL query pairs, which cover a wide range of SQL operations including SELECT, WHERE, COUNT, GROUP BY, ORDER BY, JOIN, and more. Given an English question and an SQL database scheme, the task for the model is to translate the English question into an appropriate SQL statement. Evaluation is performed via accuracy where the output is considered as correct if the model's predicted SQL query and the included GT SQL query give the same result when executed on the database. The dataset additionally categorizes each query into easy (25%), medium (40%), hard (20%), and extra hard (15%) based on the complexity of the required SQL statement. For evaluation, we report the execution accuracy of all categories.

**DART (Nan et al., 2021).**    The DART (DAta Record to Text) benchmark is a large-scale, structured dataset designed for RDF-to-text (Resource Description Framework-to-text) generation with 80,000+ instances. The DART benchmark is composed of a collection of structured data triples and corresponding text summaries which are organized into different categories. The task of the DART benchmark is to generate natural language summaries that correctly represent the given structured data inputs. DART is typically evaluated with METEOR and BLEU.

**CIFAR-10 (Krizhevsky et al., 2009).**    The CIFAR-10 (Canadian Institute For Advanced Research) dataset is a collection of images that are commonly used to train machine learning and computer vision algorithms. It is one of the most widely used datasets for image classification. The CIFAR-10 dataset contains 60,000 (50,000 for training, 10,000 for validation) $32\times32$ color images in 10 different classes. The 10 different classes are: airplane, car, bird, cat, deer, dog, frog, horse, ship, and truck. There are 6,000 images of each class. For training, we center crop each image to $24\times24$ pixels and flatten each image to a string, with a total of $24\times24\times3$ words, where each word is a number between 0-255 representing the respective pixel value. Although CIFAR-10 is a dataset for computer vision, previous work (Dinh et al., 2022) showed that Transformers can be adapted to the vision domain from the language domain. In our work, we extend this investigation to SSMs, examining their ability to perform on vision data.

**CelebA (Liu et al., 2015).**    The CelebA (CelebFaces Attributes) dataset is an extensive collection of more than 200,000 celebrity images, each tagged with 40 attributes. This dataset is notable for its diversity, volume, and comprehensive

| Dataset | | Size (Train) | Size (Val) | Size (Test) | Max. seq. len. | #Epochs | Mamba Size | Jamba Size | Metrics |
|---|---|---|---|---|---|---|---|---|---|
| GLUE | RTE | 1992 | 498 | 277 | 291 | 10 | 130M | 319M | Accuracy |
| | MRPC | 2934 | 734 | 408 | 105 | 10 | 130M | 319M | Accuracy |
| | CoLA | 6840 | 1711 | 1043 | 47 | 10 | 130M | 319M | Matthews corr. |
| | SST-2 | 53879 | 13470 | 872 | 68 | 10 | 130M | 319M | Accuracy |
| | QNLI | 83794 | 20949 | 5463 | 602 | 10 | 130M | 319M | Accuracy |
| | QQP | 291076 | 72770 | 40430 | 316 | 3 | 130M | 319M | Accuracy |
| | MNLI | 314161 | 78541 | 19647 | 425 | 3 | 130M | 319M | Accuracy |
| Spider | | 5543 | 1375 | 1034 | 1412 | 10 | 1.4B, 2.8B | 52B | Accuracy |
| SAMSum | | 14732 | 818 | 819 | 1174 | 10 | 1.4B | 52B | ROUGE |
| DART | | 62659 | 2768 | 5097 | 491 | 10 | 130M | 52B | METEOR, BLEU |
| CIFAR-10 | | 40000 | 10000 | 10000 | 1730 | 5 | 130M | 319M | Accuracy |
| CelebA | | 162770 | 19867 | 19962 | 12614 | 3 | 130M | 319M | Accuracy |

*Table 5.* **Datasets and models for our experiments.** For each dataset, we report the number of training, validation, and test samples, maximum sequence length, training epochs, model size, and evaluation metric used.

annotations, encompassing 10,177 distinct identities, 202,599 facial images, and annotations of five landmark points with 40 binary attributes per image. The dataset, which includes images with varied poses and complex backgrounds, is an essential resource for tasks in computer vision such as face recognition, attribute analysis, and detection, as well as facial landmark localization, and it offers significant utility in face editing and synthesis.

The dataset characteristics, including our train, validation and test set sizes, sequence lengths, and number of epochs, are summarized in Table 5.

## C. Details of Sec. 4: Benchmarking PEFT Methods on SSM-based Models

In this section, we provide a comprehensive experimental setup, proofs and further discussion of theoretical results, and more detailed experimental outcomes.

### C.1. Experiment Setup

For each dataset, we choose the model size depending on how challenging the dataset is and perform a small grid search for one epoch on a subset of the data (1k-2k instances) with learning rates $\{4 \times 10^{-1}, 2 \times 10^{-1}, 1 \times 10^{-1}, ..., 1 \times 10^{-5}\}$ to find the optimal learning rate of each PEFT method. We only report the validation metric of the best epoch during training (early stopping) in our results. We fine-tune pretrained Mamba and Jamba models with AdamW with a linear learning rate decay schedule. For LoRA we set rank to 8, alpha to 8, and dropout to 0.1 for all experiments. For evaluating NLG tasks, we employ beam search with five beams and a maximum beam length of 1024.

### C.2. Extended Results on Benchmarking Existing PEFT Methods

**Mamba-I.** We present comprehensive fine-tuning results for the GLUE benchmark (Wang et al., 2019), DART dataset (Nan et al., 2021), SAMSum dataset (Gliwa et al., 2019), Spider dataset (Yu et al., 2018), and CIFAR-10 (Krizhevsky et al., 2009) in Table 6, Table 7, Table 8, Table 9, and Table 10 respectively. These experimental results encompass various LoRA implementations (on different weight matrices and modules) and provide more fine-grained results across all subtasks.

**Mamba-II.** Table 11 and Table 12 present the benchmark results of LoRA and full fine-tuning across different layers of Mamba-II. We follow the same experimental setup used for Mamba-I and demonstrate that, on Mamba-II, our conclusion holds: LoRA is more effective on linear projection layers than on SSM modules.

**Jamba.** Table 13 presents the benchmark results of LoRA and full fine-tuning across different layers of Jamba. Our findings demonstrate that, on Jamba, LoRA is more effective on linear projection layers than on SSM modules, which aligns with our conclusion on Mamba.

| Layer | | Method | # Params (%) | RTE | MRPC | CoLA | SST-2 | QNLI | QQP | MNLI | Avg. |
|---|---|---|---|---|---|---|---|---|---|---|---|
| Pretrained | | | 0.00 | 46.9 | 67.9 | 0.0 | 52.4 | 50.5 | 36.8 | 32.3 | 41.0 |
| All | All | Full | 100.00 | 71.1 | 80.6 | 63.2 | 92.2 | 87.4 | 87.9 | 80.8 | 80.5 |
| | | LoRA | 1.92 | 69.9 | 80.9 | 61.4 | 91.9 | 88.4 | 87.6 | 81.1 | 80.2 |
| Prompt | Prompt Tuning | 16 tokens | 0.01 | 56.0 | 71.6 | 12.0 | 89.4 | 76.8 | 79.6 | 61.5 | 63.8 |
| | Prefix-Tuning | 1 token (no MLP) | 0.03 | 67.5 | 75.7 | 43.4 | 91.5 | 83.4 | 83.1 | 35.6 | 68.6 |
| Bias | $\beta_\Delta$, Conv1d | BitFit | 0.06 | 69.5 | 80.4 | 54.7 | 92.0 | 86.2 | 85.3 | 77.2 | 77.9 |
| Linear Projection Matrices | All | LoRA | 1.02 | 70.0 | 82.4 | 57.7 | 93.3 | 88.7 | 88.7 | 82.5 | 80.5 |
| | $W_{\text{in},x}$ | LoRA | 0.34 | 70.4 | 82.1 | 57.4 | 91.7 | 88.3 | 87.7 | 81.2 | 79.8 |
| | $W_{\text{in},z}$ | LoRA | 0.34 | 70.0 | 82.4 | 58.1 | 92.4 | 87.3 | 87.3 | 80.4 | 79.7 |
| | $W_{\text{in},x}, W_{\text{in},z}$ | LoRA | 0.68 | 70.4 | 84.3 | 62.4 | 92.5 | 88.6 | 88.3 | 81.7 | 81.2 |
| | $W_{\text{out}}$ | LoRA | 0.34 | 70.4 | 82.8 | 60.6 | 92.4 | 88.4 | 87.7 | 81.5 | 80.5 |
| S6 | All | Full | 4.31 | 69.7 | 78.9 | 59.1 | 91.5 | 88.1 | 87.5 | 80.5 | 79.3 |
| | | LoRA | 0.92 | 66.1 | 78.7 | 57.8 | 90.8 | 87.8 | 86.9 | 79.8 | 78.3 |
| | $A$ | Full | 0.46 | 68.2 | 82.1 | 54.2 | 90.9 | 86.4 | 87.9 | 79.4 | 78.4 |
| | $W_B, W_C, W_{\Delta,\downarrow}$ | Full | 2.28 | 69.7 | 77.0 | 55.8 | 91.4 | 85.4 | 85.0 | 76.8 | 77.3 |
| | | LoRA | 0.69 | 67.9 | 78.9 | 48.8 | 91.4 | 86.9 | 85.8 | 78.6 | 76.9 |
| | $W_{\Delta,\uparrow}$ | Full | 1.40 | 66.1 | 75.2 | 56.7 | 91.1 | 86.2 | 87.1 | 78.5 | 77.3 |
| | | LoRA | 0.23 | 67.1 | 79.9 | 55.1 | 90.9 | 52.7 | 86.6 | 78.7 | 73.0 |
| | Conv1d | Full | 0.14 | 68.2 | 78.4 | 57.9 | 91.1 | 86.0 | 86.0 | 78.0 | 77.9 |
| Others | $D$, LayerNorm | Full | 0.04 | 65.3 | 79.2 | 40.3 | 91.1 | 83.9 | 86.0 | 67.0 | 73.3 |

*Table 6.* **Full benchmark results on the GLUE (Wang et al., 2019) benchmark using Mamba-I 130M.** We report accuracy (↑) for RTE, MRPC, SST-2, QNLI, QQP, and MNLI tasks. CoLA performance is measured using Matthews Correlation Coefficient (↑). In each Mamba block, $W_{\text{in},x}$ and $W_{\text{in},z}$ are input projections that preprocess the input for the SSM modules and the gating branch, respectively. $W_{\text{out}}$ denotes the output projection after the gating mechanism. $W_B$ and $W_C$ are weight matrices for computing input-dependent $B_n$ and $C_n$. $W_{\Delta,\downarrow}$ and $W_{\Delta,\uparrow}$ represent down and up projections of low-rank weight matrices in the linear layer computing input-dependent step size $\Delta_n$. $\beta_\Delta$ represents the bias in this linear layer. $D$ denotes the weight of residual connections.

### C.3. Limitations of Applying Input-injection Methods on SSMs

We start by introducing the necessary notations. Denote the space of S4 mechanisms with $D$ channels as $\mathcal{F}_{S4,D}$. Let $H_0 = (h_0^{(1)}, h_0^{(2)}, \ldots, h_0^{(D)}) \in \mathbb{R}^{H \times D}$ represent the initial hidden state, and $X = (x_1, x_2, \ldots, x_N) \in \mathbb{R}^{D \times N}$ denote the input sequence. The output of the S4 mechanism is represented as $f(X; H_0)$. Furthermore, for $d$-th channel, let state transition matrix $\overline{A}^{(d)} = \text{diag}(a_1^{(d)}, \cdots, a_H^{(d)})$ and input transition vector $\overline{B}^{(d)} = (b_1, \cdots, b_H)^\top$, where $d = 1, \ldots, D$. For any vector $v \in \mathbb{R}^n$, we use $v_{i:j} \in \mathbb{R}^{j-i}$ to denote the subvector of $v$ containing elements from $i \in \mathbb{N}^+$ to $j \in \mathbb{N}^+$, where $i < j$. Similarly, for any matrix $M \in \mathbb{R}^{m \times n}$, we use $M_{i_1:j_1, i_2:j_2}$ to denote the submatrix containing rows $i_1 \in \mathbb{N}^+$ to $j_1 \in \mathbb{N}^+$ and columns $i_2 \in \mathbb{N}^+$ to $j_2 \in \mathbb{N}^+$, where $i_1 < j_1, i_2 < j_2$.

**Proposition 1** (Expressivity of Prefix-Tuning on SSMs). *Let $f \in \mathcal{F}_{S4,D}$ be an S4 mechanism. Consider prefix-tuning that prepends a sequence $P = (p_1, \ldots, p_M) \in \mathbb{R}^{D \times M}$ to the input sequence $X = (x_1, x_2, \ldots, x_N) \in \mathbb{R}^{D \times N}$. For any prefix $P \in \mathbb{R}^{D \times M}$, there exists an initial hidden state $H_0^\star \in \mathbb{R}^{H \times D}$ such that the output of S4 after prefix-tuning and that after initial state tuning are identical, i.e., $f(X; H_0^\star) \equiv f([P, X]; H_0)_{1:D, M+1:M+N}$ for all $X \in \mathbb{R}^{D \times N}$.*

*Furthermore, assume that $\prod_{0 \leq i < j \leq H} (a_j^{(d)} - a_i^{(d)}) \neq 0$ and $\prod_{k=1}^{H} b_k^{(d)} \neq 0$ for all channels $d = 1, \ldots, D$. Then the converse (i.e., for any $H_0 \in \mathbb{R}^{H \times D}$, there exists a $P^\star \in \mathbb{R}^{D \times M}$ such that $f([P^\star, X]; H_0)_{1:D, M+1:M+N} \equiv f(X; H_0^\star)$ for all $X \in \mathbb{R}^{D \times N}$) holds if and only if $M \geq H$.*

*Proof of Proposition 1.* Given that operations in S4 are independent across all channels, we can, without loss of generality, consider the case where the number of channels $D = 1$. Consequently, we can simplify our notation: the initial hidden states

| Layer | | Method | # Params (%) | METEOR | BLEU |
|---|---|---|---|---|---|
| All | All | Full | 100.00 | 71.0 | 51.8 |
| | | LoRA | 1.92 | 71.0 | 49.5 |
| | | DoRA | 2.02 | 70.9 | 51.4 |
| Prompt | Prompt Tuning | 64 tokens | 0.04 | 66.2 | 39.8 |
| | Prefix-Tuning | 64 tokens | 22.69 | 66.6 | 42.5 |
| Bias | $\boldsymbol{\beta_\Delta}$, Conv1d | BitFit | 0.06 | 67.0 | 43.7 |
| Linear Projection Matrices | All | LoRA | 1.02 | 71.2 | 49.2 |
| | | DoRA | 1.09 | 71.2 | 50.8 |
| | $\boldsymbol{W}_{\mathrm{in},x}$ | LoRA | 0.34 | 70.3 | 48.9 |
| | | DoRA | 0.37 | 70.8 | 49.9 |
| | $\boldsymbol{W}_{\mathrm{in},z}$ | LoRA | 0.34 | 70.4 | 49.1 |
| | | DoRA | 0.37 | 70.2 | 48.3 |
| | $\boldsymbol{W}_{\mathrm{in},x}, \boldsymbol{W}_{\mathrm{in},z}$ | LoRA | 0.68 | 70.9 | 49.5 |
| | | DoRA | 0.74 | 70.7 | 51.6 |
| | $\boldsymbol{W}_{\mathrm{out}}$ | LoRA | 0.34 | 70.7 | 47.0 |
| | | DoRA | 0.36 | 70.7 | 46.0 |
| S6 | All | Full | 4.31 | 70.3 | 48.7 |
| | | LoRA | 0.92 | 69.9 | 50.8 |
| | | DoRA | 0.95 | 70.2 | 50.0 |
| | $\boldsymbol{A}$ | Full | 0.46 | 69.3 | 48.1 |
| | $\boldsymbol{W}_B, \boldsymbol{W}_C, \boldsymbol{W}_{\Delta,\downarrow}$ | Full | 2.28 | 70.1 | 50.0 |
| | | LoRA | 0.69 | 68.8 | 48.0 |
| | | DoRA | 0.69 | 68.3 | 47.3 |
| | $\boldsymbol{W}_{\Delta,\uparrow}$ | Full | 1.40 | 69.6 | 47.2 |
| | | LoRA | 0.23 | 68.9 | 47.0 |
| | | DoRA | 0.26 | 68.4 | 46.3 |
| | Conv1d | Full | 0.14 | 68.6 | 47.9 |
| Others | $\boldsymbol{D}$, LayerNorm | Full | 0.04 | 67.0 | 44.2 |

*Table 7.* **Full benchmark results on the DART (Nan et al., 2021) benchmark using Mamba-I 130M.** We report METEOR ($\uparrow$) and BLEU ($\uparrow$) scores. In each Mamba block, $\boldsymbol{W}_{\mathrm{in},x}$ and $\boldsymbol{W}_{\mathrm{in},z}$ are input projections that preprocess the input for SSM modules and the gating branch, respectively. $\boldsymbol{W}_{\mathrm{out}}$ denotes the output projection after the gating mechanism. $\boldsymbol{W}_B$ and $\boldsymbol{W}_C$ are weight matrices for computing input-dependent $\boldsymbol{B}_n$ and $\boldsymbol{C}_n$. $\boldsymbol{W}_{\Delta,\downarrow}$ and $\boldsymbol{W}_{\Delta,\uparrow}$ represent down and up projections of low-rank weight matrices in the linear layer computing input-dependent step size $\boldsymbol{\Delta}_n$. $\boldsymbol{\beta_\Delta}$ represents the bias in this linear layer. $\boldsymbol{D}$ denotes the weight of residual connections.

| Layer | | Method | # Params (%) | R1 | R2 | RL |
|---|---|---|---|---|---|---|
| All | All | Full | 100.00 | 51.2 | 27.3 | 42.9 |
| | | LoRA | 0.97 | 50.8 | 26.6 | 42.7 |
| Prompt | Prompt Tuning | 64 tokens | 0.01 | 50.1 | 25.6 | 41.6 |
| | Prefix-Tuning | 64 tokens | 12.81 | 50.6 | 26.5 | 42.1 |
| Bias | $\boldsymbol{\beta_\Delta}$, Conv1d | BitFit | 0.03 | 50.3 | 25.7 | 41.9 |
| Linear Projection Matrices | All | LoRA | 0.51 | 50.8 | 26.9 | 42.8 |
| | $\boldsymbol{W}_{\mathrm{in},x}$ | LoRA | 0.17 | 49.8 | 25.4 | 41.2 |
| | $\boldsymbol{W}_{\mathrm{in},z}$ | LoRA | 0.17 | 50.0 | 26.1 | 41.7 |
| | $\boldsymbol{W}_{\mathrm{in},x}, \boldsymbol{W}_{\mathrm{in},z}$ | LoRA | 0.34 | 50.9 | 27.0 | 42.3 |
| | $\boldsymbol{W}_{\mathrm{out}}$ | LoRA | 0.17 | 49.9 | 25.4 | 41.5 |
| S6 | All | Full | 4.46 | 51.1 | 26.9 | 42.2 |
| | | LoRA | 0.46 | 50.5 | 26.4 | 42.2 |
| | $\boldsymbol{A}$ | Full | 0.23 | 50.1 | 25.9 | 41.7 |
| | $\boldsymbol{W}_B, \boldsymbol{W}_C, \boldsymbol{W}_{\Delta,\downarrow}$ | Full | 2.29 | 50.5 | 26.0 | 41.8 |
| | | LoRA | 0.35 | 50.4 | 26.0 | 41.8 |
| | $\boldsymbol{W}_{\Delta,\uparrow}$ | Full | 1.85 | 50.3 | 25.7 | 41.6 |
| | | LoRA | 0.12 | 50.2 | 25.4 | 41.3 |
| | Conv1d | Full | 0.07 | 50.1 | 25.7 | 41.9 |
| Others | $\boldsymbol{D}$, LayerNorm | Full | 0.02 | 49.6 | 24.8 | 41.1 |

*Table 8.* **Full benchmark results on the SAMSum (Gliwa et al., 2019) benchmark using Mamba-I 1.4B.** R1, R2, and RL represent ROUGE-1 ($\uparrow$), ROUGE-2 ($\uparrow$), and ROUGE-L ($\uparrow$), respectively. In each Mamba block, $\boldsymbol{W}_{\mathrm{in},x}$ and $\boldsymbol{W}_{\mathrm{in},z}$ are input projections that preprocess the input for SSM modules and the gating branch, respectively. $\boldsymbol{W}_{\mathrm{out}}$ denotes the output projection after the gating mechanism. $\boldsymbol{W}_B$ and $\boldsymbol{W}_C$ are weight matrices for computing input-dependent $\boldsymbol{B}_n$ and $\boldsymbol{C}_n$. $\boldsymbol{W}_{\Delta,\downarrow}$ and $\boldsymbol{W}_{\Delta,\uparrow}$ represent down and up projections of low-rank weight matrices in the linear layer computing input-dependent step size $\boldsymbol{\Delta}_n$. $\boldsymbol{\beta_\Delta}$ represents the bias in this linear layer. $\boldsymbol{D}$ denotes the weight of residual connections.

| Layer | | Method | # Params (%) | All | Easy | Medium | Hard | Extra |
|---|---|---|---|---|---|---|---|---|
| All | All | Full | 100.00 | 66.2 | 84.3 | 69.5 | 53.4 | 43.4 |
| | | LoRA | 0.97 | 56.4 | 76.2 | 57.0 | 47.7 | 34.3 |
| | | DoRA | 1.02 | 55.7 | 77.0 | 57.0 | 47.1 | 29.5 |
| Prompt | Prompt Tuning | 64 tokens | 0.01 | 43.6 | 65.3 | 42.4 | 33.3 | 25.3 |
| | Prefix-Tuning | 64 tokens | 12.81 | 39.7 | 65.7 | 38.6 | 31.0 | 15.1 |
| Bias | $\beta_{\mathbf{\Delta}}$,Conv1d | BitFit | 0.03 | 51.3 | 74.2 | 50.9 | 43.1 | 26.5 |
| Linear Projection Matrices | All | LoRA | 0.51 | 54.7 | 75.0 | 55.6 | 46.0 | 31.3 |
| | | DoRA | 0.55 | 57.2 | 79.4 | 58.7 | 46.0 | 31.3 |
| | $\mathbf{W}_{\mathrm{in},x}$ | LoRA | 0.17 | 60.8 | 76.6 | 63.5 | 52.9 | 38.6 |
| | | DoRA | 0.19 | 58.4 | 80.2 | 60.1 | 49.4 | 30.7 |
| | $\mathbf{W}_{\mathrm{in},z}$ | LoRA | 0.17 | 46.3 | 68.5 | 45.7 | 36.8 | 24.7 |
| | | DoRA | 0.19 | 59.8 | 83.9 | 60.1 | 50.6 | 32.5 |
| | $\mathbf{W}_{\mathrm{in},x}, \mathbf{W}_{\mathrm{in},z}$ | LoRA | 0.34 | 57.5 | 77.4 | 58.7 | 45.4 | 37.3 |
| | | DoRA | 0.37 | 60.7 | 78.6 | 62.1 | 52.9 | 38.6 |
| | $\mathbf{W}_{\mathrm{out}}$ | LoRA | 0.17 | 61.8 | 81.9 | 65.2 | 45.4 | 39.8 |
| | | DoRA | 0.18 | 61.3 | 79.4 | 63.9 | 50.0 | 39.2 |
| S6 | All | Full | 4.46 | 56.7 | 76.6 | 57.8 | 46.0 | 34.9 |
| | | LoRA | 0.46 | 56.3 | 75.0 | 56.5 | 50.6 | 33.7 |
| | | DoRA | 0.48 | 58.9 | 77.4 | 62.1 | 47.1 | 34.9 |
| | $\mathbf{A}$ | Full | 0.23 | 51.1 | 71.4 | 52.5 | 42.5 | 25.9 |
| | $\mathbf{W}_B, \mathbf{W}_C, \mathbf{W}_{\mathbf{\Delta},\downarrow}$ | Full | 2.29 | 47.2 | 72.2 | 46.9 | 35.6 | 22.9 |
| | | LoRA | 0.35 | 55.0 | 73.8 | 56.7 | 44.3 | 33.7 |
| | | DoRA | 0.35 | 55.3 | 78.2 | 57.8 | 41.4 | 28.9 |
| | $\mathbf{W}_{\mathbf{\Delta},\uparrow}$ | Full | 1.85 | 56.8 | 77.0 | 59.4 | 43.7 | 33.1 |
| | | LoRA | 0.12 | 58.0 | 78.6 | 59.4 | 48.9 | 33.1 |
| | | DoRA | 0.13 | 55.3 | 76.2 | 59.2 | 42.5 | 27.1 |
| | Conv1d | Full | 0.07 | 53.2 | 74.6 | 52.9 | 43.7 | 31.9 |
| Others | $\mathbf{D}$,LayerNorm | Full | 0.02 | 49.6 | 70.6 | 50.4 | 40.2 | 25.9 |

(a) Full benchmark results on Spider using Mamba-I 1.4B.

| Layer | | Method | # Params (%) | All | Easy | Medium | Hard | Extra |
|---|---|---|---|---|---|---|---|---|
| All | All | Full | 100.00 | 71.8 | 87.5 | 73.5 | 63.8 | 51.8 |
| | | LoRA | 0.80 | 70.9 | 90.7 | 74.0 | 58.6 | 45.8 |
| Prompt | Prompt Tuning | 64 tokens | 0.01 | 50.7 | 75.4 | 53.8 | 37.4 | 19.3 |
| | Prefix-Tuning | 1 token | 10.82 | 45.1 | 75.0 | 45.1 | 32.2 | 13.9 |
| Bias | $\beta_{\mathbf{\Delta}}$,Conv1d | BitFit | 0.02 | 59.9 | 82.3 | 60.8 | 52.9 | 31.3 |
| Linear Projection Matrices | All | LoRA | 0.42 | 58.2 | 74.6 | 58.3 | 51.7 | 40.4 |
| | $\mathbf{W}_{\mathrm{in},x}$ | LoRA | 0.14 | 66.7 | 87.9 | 67.7 | 56.9 | 42.8 |
| | $\mathbf{W}_{\mathrm{in},z}$ | LoRA | 0.14 | 65.4 | 86.7 | 68.8 | 54.6 | 35.5 |
| | $\mathbf{W}_{\mathrm{in},x}, \mathbf{W}_{\mathrm{in},z}$ | LoRA | 0.28 | 65.2 | 89.1 | 67.3 | 51.7 | 38.0 |
| | $\mathbf{W}_{\mathrm{out}}$ | LoRA | 0.14 | 67.0 | 87.1 | 69.1 | 52.9 | 46.4 |
| S6 | All | Full | 4.44 | 65.7 | 81.9 | 68.8 | 58.0 | 41.0 |
| | | LoRA | 0.38 | 63.9 | 86.3 | 68.2 | 49.4 | 34.3 |
| | $\mathbf{A}$ | Full | 0.19 | 56.6 | 77.0 | 58.1 | 46.0 | 33.1 |
| | $\mathbf{W}_B, \mathbf{W}_C, \mathbf{W}_{\mathbf{\Delta},\downarrow}$ | Full | 2.27 | 58.8 | 79.0 | 61.0 | 50.6 | 31.3 |
| | | LoRA | 0.29 | 60.3 | 82.7 | 63.0 | 46.6 | 33.7 |
| | $\mathbf{W}_{\mathbf{\Delta},\uparrow}$ | Full | 1.91 | 62.2 | 82.3 | 65.7 | 51.7 | 33.7 |
| | | LoRA | 0.10 | 62.2 | 80.2 | 66.6 | 49.4 | 36.7 |
| | Conv1d | Full | 0.06 | 62.5 | 81.9 | 66.1 | 51.1 | 35.5 |
| Others | $\mathbf{D}$,LayerNorm | Full | 0.02 | 51.0 | 71.0 | 51.1 | 42.5 | 29.5 |

(b) Full benchmark results on Spider using Mamba-I 2.8B.

*Table 9.* **Full benchmark results on Spider** (Yu et al., 2018) **dataset using Mamba-I.** We report the accuracy (↑) for Spider and its subsets. We consider two models in our experiments: Mamba-I 1.4B and Mamba-I 2.8B. In each Mamba block, $\mathbf{W}_{\mathrm{in},x}$ and $\mathbf{W}_{\mathrm{in},z}$ are input projections that preprocess the input for SSM modules and the gating branch, respectively. $\mathbf{W}_{\mathrm{out}}$ denotes the output projection after the gating mechanism. $\mathbf{W}_B$ and $\mathbf{W}_C$ are weight matrices for computing input-dependent $\mathbf{B}_n$ and $\mathbf{C}_n$. $\mathbf{W}_{\mathbf{\Delta},\downarrow}$ and $\mathbf{W}_{\mathbf{\Delta},\uparrow}$ represent down and up projections of low-rank weight matrices in the linear layer computing input-dependent step size $\mathbf{\Delta}_n$. $\beta_{\mathbf{\Delta}}$ represents the bias in this linear layer. $\mathbf{D}$ denotes the weight of residual connections.

| Layer | | Method | # Params (%) | Accuracy |
|---|---|---|---|---|
| Pretrained | | | 0.00 | 0.08 |
| All | All | Full | 100.00 | 59.96 |
| | | LoRA | 1.92 | 60.35 |
| Bias | $\boldsymbol{\beta_{\Delta}}$, `Conv1d` | BitFit | 0.06 | 44.40 |
| Linear Projection Matrices | All | LoRA | 1.02 | 62.79 |
| | $\boldsymbol{W}_{\text{in},x}$ | LoRA | 0.34 | 53.49 |
| | $\boldsymbol{W}_{\text{in},z}$ | LoRA | 0.34 | 58.15 |
| | $\boldsymbol{W}_{\text{in},x}, \boldsymbol{W}_{\text{in},z}$ | LoRA | 0.68 | 61.04 |
| | $\boldsymbol{W}_{\text{out}}$ | LoRA | 0.34 | 52.04 |
| S6 | All | Full | 4.31 | 55.51 |
| | | LoRA | 0.92 | 43.96 |
| | $\boldsymbol{A}$ | Full | 0.46 | 61.21 |
| | $\boldsymbol{W_B}, \boldsymbol{W_C}, \boldsymbol{W_{\Delta,\downarrow}}$ | Full | 2.28 | 49.51 |
| | | LoRA | 0.69 | 52.27 |
| | $\boldsymbol{W_{\Delta,\uparrow}}$ | Full | 1.40 | 34.54 |
| | | LoRA | 0.23 | 56.49 |
| | `Conv1d` | Full | 0.14 | 55.65 |
| Others | $\boldsymbol{D}$, `LayerNorm` | Full | 0.04 | 58.09 |

*Table 10.* **Full benchmark results on the CIFAR-10 (Krizhevsky et al., 2009) dataset using Mamba-I 130M.** We report accuracy ($\uparrow$). In each Mamba block, $\boldsymbol{W}_{\text{in},x}$ and $\boldsymbol{W}_{\text{in},z}$ are input projections that preprocess the input for SSM modules and the gating branch, respectively. $\boldsymbol{W}_{\text{out}}$ denotes the output projection after the gating mechanism. $\boldsymbol{W_B}$ and $\boldsymbol{W_C}$ are weight matrices for computing input-dependent $\boldsymbol{B}_n$ and $\boldsymbol{C}_n$. $\boldsymbol{W_{\Delta,\downarrow}}$ and $\boldsymbol{W_{\Delta,\uparrow}}$ represent down and up projections of low-rank weight matrices in the linear layer computing input-dependent step size $\boldsymbol{\Delta}_n$. $\boldsymbol{\beta_{\Delta}}$ represents the bias in this linear layer. $\boldsymbol{D}$ denotes the weight of residual connections.

$\boldsymbol{H}_0 \in \mathbb{R}^{H \times D}$ become $\boldsymbol{h}_0 \in \mathbb{R}^H$, the input sequence $\boldsymbol{X} \in \mathbb{R}^{D \times N}$ becomes $\boldsymbol{x} \in \mathbb{R}^N$, and the prefix $\boldsymbol{P} \in \mathbb{R}^{D \times M}$ becomes $\boldsymbol{p} \in \mathbb{R}^M$. We omit the superscript $(d)$ denoting the channel index. To differentiate between the hidden states and output of prefix-tuned S4 (i.e., $f([\boldsymbol{P}, \boldsymbol{X}]; \boldsymbol{H}_0)_{1:D, M+1:M+N}$) and initial state tuned S4 (i.e., $f(\boldsymbol{X}; \boldsymbol{H}_0^{\star})$), we introduce superscripts "PT" and "IST" respectively. The "PT" superscript denotes hidden states and output of S4 after prefix-tuning, while "IST" indicates those after initial state tuning.

We divide the proposition into two statements:

1. For any prefix $\boldsymbol{p} \in \mathbb{R}^M$, there exists an initial hidden state $\boldsymbol{h}_0^{\star} \in \mathbb{R}^H$ such that the output of S4 after prefix-tuning and that after initial state tuning are identical, i.e., $f(\boldsymbol{x}; \boldsymbol{h}_0^{\star}) \equiv f([\boldsymbol{p}, \boldsymbol{x}]; \boldsymbol{h}_0)_{M+1:N+M}$ for all $\boldsymbol{x} \in \mathbb{R}^N$.

2. Furthermore, assume that $\prod_{0 \le i < j \le H}(a_j - a_i) \ne 0$ and $\prod_{k=1}^H b_k \ne 0$. Then the converse (i.e., for any $\boldsymbol{h}_0 \in \mathbb{R}^H$, there exists a $\boldsymbol{p}^{\star} \in \mathbb{R}^M$ such that $f([\boldsymbol{p}^{\star}, \boldsymbol{x}]; \boldsymbol{h}_0)_{M+1:N+M} \equiv f(\boldsymbol{x}; \boldsymbol{h}_0^{\star})$ for all $\boldsymbol{x} \in \mathbb{R}^N$) holds if and only if $M \ge H$.

We will first prove the first statement and then proceed to prove the second statement.

**Statement 1.** The recurrent computation formulation of S4 in (2) implies that for each position $i$, the output $y_i$ depends solely on the previous hidden state $h_{i-1}$ and the current input $x_i$. Thus, to demonstrate that $f(\boldsymbol{x}; \boldsymbol{h}_0^{\star}) \equiv f([\boldsymbol{p}, \boldsymbol{x}]; \boldsymbol{h}_0)_{M+1:N+M}$ for all $\boldsymbol{x} \in \mathbb{R}^N$, it suffices to show that the hidden state for predicting output $y_1^{\text{IST}}$ equals that for predicting output $y_{M+1}^{\text{PT}}$, where $y_1^{\text{IST}}$ and $y_{M+1}^{\text{PT}}$ are outputs corresponding to the input $x_1$ for initial state tuning and prefix-tuning, respectively. In other words, it is sufficient to show that the initial state of initial-state-tuned model $\boldsymbol{h}_0^{\text{IST}} = \boldsymbol{h}_0^{\star}$ is equal to the $(M+1)$-th hidden state of prefix-tuned model $\boldsymbol{h}_{M+1}^{\text{PT}} = \sum_{m=1}^M \overline{\boldsymbol{A}}^{M-m} \overline{\boldsymbol{B}} p_m$. When this equality holds, the subsequent hidden states and outputs for both versions of S4 will be identical, as the input sequence from that point onward is the same. Therefore,

| Layer | | Method | # Params (%) | METEOR | BLEU |
|---|---|---|---|---|---|
| All | All | Full | 100.00 | 66.6 | 34.9 |
| | | LoRA | 1.39 | 66.9 | 45.4 |
| Linear Projection Matrices | $\boldsymbol{W}_{\text{in}}, \boldsymbol{W}_{\text{out}}$ | LoRA | 1.02 | 67.1 | 44.7 |
| | $\boldsymbol{W}_{\text{in}}$ | LoRA | 0.68 | 67.1 | 43.0 |
| | $\boldsymbol{W}_{\text{out}}$ | LoRA | 0.34 | 66.8 | 42.3 |
| S6 | All | Full | 4.17 | 65.7 | 39.7 |
| | | LoRA | 0.38 | 64.2 | 40.1 |
| | $\boldsymbol{W}_B, \boldsymbol{W}_C, \boldsymbol{W}_\Delta$ | Full | 4.00 | 66.0 | 36.2 |
| | | LoRA | 0.38 | 64.8 | 39.5 |

*Table 11.* **Full benchmark results of LoRA on DART ([Nan et al., 2021](#)) dataset using Mamba-II 130M.**

| Layer | | Method | # Params (%) | All | Easy | Medium | Hard | Extra |
|---|---|---|---|---|---|---|---|---|
| All | All | Full | 100.00 | 64.8 | 85.9 | 65.7 | 54.0 | 42.2 |
| | | LoRA | 0.71 | 64.5 | 81.0 | 66.4 | 56.9 | 42.8 |
| Linear Projection Matrices | $\boldsymbol{W}_{\text{in}}, \boldsymbol{W}_{\text{out}}$ | LoRA | 0.52 | 50.4 | 68.5 | 52.0 | 44.8 | 24.7 |
| | $\boldsymbol{W}_{\text{in}}$ | LoRA | 0.35 | 57.5 | 76.2 | 59.4 | 48.9 | 33.7 |
| | $\boldsymbol{W}_{\text{out}}$ | LoRA | 0.18 | 57.9 | 81.0 | 56.7 | 51.7 | 33.1 |
| S6 | All | Full | 2.42 | 55.1 | 76.2 | 56.1 | 42.5 | 34.3 |
| | | LoRA | 0.18 | 54.1 | 74.2 | 58.1 | 46.0 | 21.7 |
| | $\boldsymbol{A}_{\text{log}}$ | Full | 0.00 | 21.5 | 46.0 | 18.8 | 11.5 | 2.4 |
| | $\boldsymbol{W}_B, \boldsymbol{W}_C, \boldsymbol{W}_\Delta$ | Full | 2.34 | 50.3 | 73.0 | 52.2 | 39.7 | 22.3 |
| | | LoRA | 0.18 | 55.5 | 77.4 | 55.2 | 46.6 | 33.1 |

*Table 12.* **Full benchmark results on the Spider ([Yu et al., 2018](#)) dataset using Mamba-II 1.3B.**

we prove the first statement by letting

$$\boldsymbol{h}_0^\star = \sum_{m=1}^{M} \overline{\boldsymbol{A}}^{M-m} \overline{\boldsymbol{B}} p_m.$$

**Statement 2.** We aim to investigate the conditions under which there exists a $\boldsymbol{h}_0^\star \in \mathbb{R}^H$ such that for any $\boldsymbol{p} \in \mathbb{R}^M$, $f([\boldsymbol{p}^\star, \boldsymbol{x}]; \boldsymbol{h}_0)_{M+1:N+M} \neq f(\boldsymbol{x}; \boldsymbol{h}_0^\star)$. This is equivalent to demonstrating the existence of $\boldsymbol{h}_0^\star \in \mathbb{R}^H$ such that

$$\boldsymbol{h}_0^\star \neq \sum_{m=1}^{M} \overline{\boldsymbol{A}}^{M-m} \overline{\boldsymbol{B}} p_m, \quad \text{for all } \boldsymbol{p} \in \mathbb{R}^M.$$

This condition can be further reformulated as

$$\mathbb{R}^H \setminus \text{span}(\overline{\boldsymbol{A}}^M \overline{\boldsymbol{B}}, \overline{\boldsymbol{A}}^{M-1} \overline{\boldsymbol{B}}, \ldots, \overline{\boldsymbol{B}}) \neq \emptyset,$$

which is equivalent to

$$\text{span}(\overline{\boldsymbol{A}}^M \overline{\boldsymbol{B}}, \overline{\boldsymbol{A}}^{M-1} \overline{\boldsymbol{B}}, \ldots, \overline{\boldsymbol{B}}) \subsetneq \mathbb{R}^H. \tag{6}$$

To determine when this condition holds, we analyze three distinct cases: (i) $M < H$, (ii) $M = H$, and (iii) $M > H$.

*(Case 1: When $M < H$).* In this scenario, it is obvious that (6) holds. The existence of such a $\boldsymbol{h}_0^\star$ is guaranteed because the dimension of the span is at most $M$, which is strictly less than $H$. This choice of $\boldsymbol{h}_0^\star$ ensures that it cannot be represented as a linear combination of the vectors in the span, thereby establishing the inequality.

*(Case 2: When $M = H$).* In this scenario, $\text{span}(\overline{\boldsymbol{A}}^M \overline{\boldsymbol{B}}, \overline{\boldsymbol{A}}^{M-1} \overline{\boldsymbol{B}}, \ldots, \overline{\boldsymbol{B}}) = \mathbb{R}^H$ if and only if $(\overline{\boldsymbol{A}}^M \overline{\boldsymbol{B}}, \overline{\boldsymbol{A}}^{M-1} \overline{\boldsymbol{B}}, \ldots, \overline{\boldsymbol{B}})$ are linearly independent. Note that

$$\det(\overline{\boldsymbol{A}}^M \overline{\boldsymbol{B}}, \overline{\boldsymbol{A}}^{M-1} \overline{\boldsymbol{B}}, \ldots, \overline{\boldsymbol{B}}) = \det(\overline{\boldsymbol{A}}^M, \overline{\boldsymbol{A}}^{M-1}, \ldots, \mathbf{1}) \prod_{k=1}^{H} b_k, \tag{7}$$

| Layer | | Method | # Params (%) | METEOR | BLEU |
|---|---|---|---|---|---|
| All | All | Full | 100.00 | 70.8 | 45.0 |
| Attention | All | LoRA | 0.02 | 63.5 | 19.7 |
| MLP | All | LoRA | 1.37 | 70.9 | 46.2 |
| Linear Projection Matrices + S6 | All | LoRA | 0.31 | 70.2 | 40.0 |
| Linear Projection Matrices | $W_{in}$ | LoRA | 0.11 | 68.9 | 37.8 |
| | $W_{out}$ | LoRA | 0.05 | 67.7 | 31.9 |
| S6 | All | Full | 0.54 | 69.2 | 35.5 |
| | $W_B, W_C, W_{\Delta,\downarrow}$ | LoRA | 0.15 | 66.6 | 24.2 |

*Table 13.* **Full benchmark results on DART (Nan et al., 2021) dataset using Jamba-Tiny-319M.**

| Task | RTE | MRPC | CoLA | SST-2 | QNLI | QQP | MNLI | Avg. Score |
|---|---|---|---|---|---|---|---|---|
| Prompt Tuning | 56.0 | 71.6 | 12.0 | 89.4 | 76.8 | 79.6 | 61.5 | 63.8 |
| Prefix-Tuning | 69.5 | 75.7 | 43.4 | 91.5 | 83.4 | 83.1 | 35.6 | 68.6 |
| Initial State Tuning | 66.8 | 75.1 | 52.4 | 92.4 | 86.4 | 86.1 | 78.5 | 76.8 |
| LoRA (Linear Projection Matrices) | **70.4** | **82.8** | **60.6** | **92.4** | **88.4** | **87.7** | **81.5** | **80.5** |

*Table 14.* **Comparison of prompt-tuning, prefix-tuning, initial state tuning, and LoRA on seven tasks from the GLUE benchmark.**
We report Matthews correlation ($\uparrow$) for CoLA, overall (matched and mismatched) accuracy ($\uparrow$) for MNLI, and accuracy for other tasks.
Initial state tuning and LoRA are constrained to use less than 0.5% trainable parameters. **Bold** numbers indicate the best performance
across all three methods, while underlined numbers show the highest score among input-injection methods (prefix-tuning and initial state
tuning). Initial state tuning outperforms prefix-tuning and prompt-tuning on five out of seven tasks, while LoRA consistently outperforms
all input-injection methods.

where

$$\det(\overline{\boldsymbol{A}}^M, \overline{\boldsymbol{A}}^{M-1}, \ldots, \mathbf{1}) = \det \begin{bmatrix} a_1^{H-1} & \cdots & a_1^2 & a_1 & 1 \\ a_2^{H-1} & \cdots & a_2^2 & a_2 & 1 \\ \vdots & \ddots & \vdots & \vdots & \vdots \\ a_H^{H-1} & \cdots & a_H^2 & a_H & 1 \end{bmatrix} \qquad \text{(Expand)}$$

$$= (-1)^{\frac{H(H-1)}{2}} \prod_{0 \le i < j \le H}^{H} (a_j - a_i). \qquad \text{(Vandermonde matrix)} \qquad (8)$$

Combining (7) and (8) yields

$$\det(\overline{\boldsymbol{A}}^M \overline{\boldsymbol{B}}, \overline{\boldsymbol{A}}^{M-1} \overline{\boldsymbol{B}}, \ldots, \overline{\boldsymbol{B}}) = (-1)^{\frac{H(H-1)}{2}} \prod_{0 \le i < j \le H}^{H} (a_j - a_i) \prod_{k=1}^{H} b_k.$$

Therefore, if and only if $\prod_{1 \le i < j \le H} (a_j - a_i) \ne 0$ and $\prod_{k=1}^{H} b_k \ne 0$, we have

$$\det(\overline{\boldsymbol{A}}^M \overline{\boldsymbol{B}}, \overline{\boldsymbol{A}}^{M-1} \overline{\boldsymbol{B}}, \ldots, \overline{\boldsymbol{B}}) \ne 0,$$

which is both necessary and sufficient for the linear independence of $(\overline{\boldsymbol{A}}^M \overline{\boldsymbol{B}}, \overline{\boldsymbol{A}}^{M-1} \overline{\boldsymbol{B}}, \ldots, \overline{\boldsymbol{B}})$, and consequently, for the
condition in (6) to be satisfied.

*(Case 3: When $M > H$).* The analysis presented in case 2 extends naturally to this scenario.

The combination of the three cases above completes the proof of statement 2. $\qquad\square$

### C.4. Optimal Application of LoRA$^\star$ in SSM-based Models

Several studies (Hu et al., 2023; He et al., 2021) present findings on Transformers, indicating that applying LoRA$^\star$ to linear
projection matrices yields performance comparable to or marginally superior to that of attention layers. In contrast, our
experimental results on SSMs reveal that applying LoRA$^\star$ to linear projection matrices is more effective than applying it to
S6. To elucidate this phenomenon, we examine the influence of updating linear projection matrices on the model's output.

**Notations.** To make the analysis tractable, we consider a simplified SSM-based architecture composed of the following components:

- Two input projection matrices: $\boldsymbol{W}_{\text{in},1}, \boldsymbol{W}_{\text{in},2} \in \mathbb{R}^{D \times D}$;

- The S6 module parameterized by diagonal state transition matrices $\{\boldsymbol{A}^{(d)}\}_{d=1}^{D}$ with $\boldsymbol{A}^{(d)} \in \mathbb{R}^{H \times H}$, the weight matrices $\boldsymbol{W}_B, \boldsymbol{W}_C \in \mathbb{R}^{H \times D}$ for computing input-dependent input transition vectors $\boldsymbol{B}_n \in \mathbb{R}^H$ and output mapping vectors $\boldsymbol{C}_n \in \mathbb{R}^H$, the down and up projection matrices $\boldsymbol{W}_{\boldsymbol{\Delta},\downarrow} \in \mathbb{R}^{D \times R}, \boldsymbol{W}_{\boldsymbol{\Delta},\uparrow} \in \mathbb{R}^{R \times D}$ (where $R$ is the rank) for low-rank weight matrices for computing the input-dependent step size $\boldsymbol{\Delta}_n = (\Delta_n^{(1)}, \ldots, \Delta_n^{(D)}) \in \mathbb{R}_D$, for $n = 1, \ldots, N$.

Define $\boldsymbol{W}_{\text{S6}} = [\boldsymbol{W}_B^\top, \boldsymbol{W}_C^\top, \boldsymbol{W}_{\boldsymbol{\Delta},\uparrow}^\top]^\top \in \mathbb{R}^{(2H+R) \times D}$. In the Mamba implementation, $\boldsymbol{W}_{\text{S6}}$ is implemented as the weight matrix of a single linear layer, referred to as `x_proj` in the codebase. Let the input sequence be $\boldsymbol{X} = (\boldsymbol{x}_1, \ldots, \boldsymbol{x}_N) \in \mathbb{R}^{D \times N}$. At each time step $n$, the S6 module uses two differently projected versions of the input: (i) the input projected via $\boldsymbol{W}_{\text{in},1}$ is used to compute the input-dependent parameters $\overline{\boldsymbol{A}}_n$, $\overline{\boldsymbol{B}}_n$, and $\boldsymbol{C}_n$, and (ii) the input projected via $\boldsymbol{W}_{\text{in},2}$ serves as the actual input to the S6 module. We note that this formulation generalizes the standard case, which uses a single input projection matrix before the S6 module. In particular, it reduces to the standard case when $\boldsymbol{W}_{\text{in},1} = \boldsymbol{W}_{\text{in},2}$. Then, the output at time step $N$ is given by:

$$y_N^{(d)} = \overbrace{\boldsymbol{C}(\boldsymbol{W}_{\text{in},1}\boldsymbol{x}_N)^\top}^{\text{Input-dependent } \boldsymbol{C}_N} \sum_{n=1}^{N} \underbrace{\left( \prod_{m=1}^{n} \overbrace{\overline{\boldsymbol{A}}(\boldsymbol{W}_{\text{in},1}\boldsymbol{x}_m)}^{\text{Input-dependent } \overline{\boldsymbol{A}}_m} \right) \overbrace{\overline{\boldsymbol{B}}_n(\boldsymbol{W}_{\text{in},1}\boldsymbol{x}_n)}^{\text{Input-dependent } \overline{\boldsymbol{B}}_n}}_{\text{Parameters depending on input after projection } \boldsymbol{W}_{\text{in},1}} \underbrace{(\boldsymbol{W}_{\text{in},2}\boldsymbol{x}_n)^{(d)}}_{\text{Input after projection } \boldsymbol{W}_{\text{in},2}}.$$

To be more specific, the definitions for the relevant terms are:

$$\boldsymbol{\Delta}_n = \text{softplus}(\boldsymbol{W}_{\boldsymbol{\Delta},\downarrow}\boldsymbol{W}_{\boldsymbol{\Delta},\uparrow}\boldsymbol{W}_{\text{in},1}\boldsymbol{x}_n + \boldsymbol{\beta}_{\boldsymbol{\Delta}}), \tag{9}$$

$$\overline{\boldsymbol{A}}_n^{(d)} = \exp(\Delta_n^{(d)}\boldsymbol{A}^{(d)}), \quad \overline{\boldsymbol{B}}_n^{(d)} = \Delta_n^{(d)}\boldsymbol{W}_B\boldsymbol{W}_{\text{in},1}\boldsymbol{x}_n, \quad \boldsymbol{C}_n = \boldsymbol{W}_C\boldsymbol{W}_{\text{in},1}\boldsymbol{x}_n.$$

When $\boldsymbol{\beta}_{\boldsymbol{\Delta}} = \boldsymbol{0}$, the output at time step $N$ can be further written as

$$y_N^{(d)} = (\boldsymbol{W}_C\boldsymbol{W}_{\text{in},1}\boldsymbol{x}_N)^\top \sum_{n=1}^{N} \left( \prod_{m=1}^{n} \exp(\Delta_n^{(d)}\boldsymbol{A}^{(d)}) \right) \Delta_n^{(d)}\boldsymbol{W}_B\boldsymbol{W}_{\text{in},1}\boldsymbol{x}_n(\boldsymbol{W}_{\text{in},2}^\star\boldsymbol{x}_n)^{(d)}, \tag{10}$$

$$\text{where } \boldsymbol{\Delta}_n = \text{softplus}(\boldsymbol{W}_{\boldsymbol{\Delta},\downarrow}\boldsymbol{W}_{\boldsymbol{\Delta},\uparrow}\boldsymbol{W}_{\text{in},1}\boldsymbol{x}_n).$$

**Theoretical Analysis.** Assume none of the parameters are zero and $D > 2H + R$, where $R$ is the rank of $\boldsymbol{W}_{\boldsymbol{\Delta},\downarrow}\boldsymbol{W}_{\boldsymbol{\Delta},\uparrow}$. Lemma 1 in the main body shows that applying LoRA$^\star$ solely to $\boldsymbol{W}_{\text{in},1}$ is equivalent to applying it to $\boldsymbol{W}_{\text{S6}}$. For completeness, we provide the proof below and restate the lemma for the reader's convenience.

**Lemma 1** (Expressivity of Fine-Tuning Projection Matrices). *Consider two models with the architecture described above. Let:*

- *A target model $f^\star$ parameterized by $(\{\boldsymbol{A}^{\star(d)}\}_{d=1}^{D}, \boldsymbol{W}_B^\star, \boldsymbol{W}_C^\star, \boldsymbol{W}_{\boldsymbol{\Delta},\uparrow}^\star, \boldsymbol{W}_{\boldsymbol{\Delta},\downarrow}^\star, \boldsymbol{W}_{in,1}^\star, \boldsymbol{W}_{in,2}^\star)$;*

- *A frozen model $f_0$ parameterized by $(\{\boldsymbol{A}^{\star(d)}\}_{d=1}^{D}, \boldsymbol{W}_B, \boldsymbol{W}_C, \boldsymbol{W}_{\boldsymbol{\Delta},\uparrow}, \boldsymbol{W}_{\boldsymbol{\Delta},\downarrow}^\star, \boldsymbol{W}_{in,1}, \boldsymbol{W}_{in,2}^\star)$.*

*The two models share $\{\boldsymbol{A}^{\star(d)}\}_{d=1}^{D}$, $\boldsymbol{W}_{\boldsymbol{\Delta},\downarrow}^\star$, and $\boldsymbol{W}_{in,2}^\star$, while differing in $\boldsymbol{W}_B$, $\boldsymbol{W}_C$, $\boldsymbol{W}_{\boldsymbol{\Delta},\uparrow}$, and $\boldsymbol{W}_{in,1}$. Then, there exists a projection matrix $\widehat{\boldsymbol{W}}_{in,1}$ such that the frozen model matches the output of the target model for any input sequence, i.e.,*

$$f(\cdot; \{\boldsymbol{A}^{\star(d)}\}_{d=1}^{D}, \boldsymbol{W}_B, \boldsymbol{W}_C, \boldsymbol{W}_{\boldsymbol{\Delta},\uparrow}, \boldsymbol{W}_{\boldsymbol{\Delta},\downarrow}^\star, \widehat{\boldsymbol{W}}_{in,1}, \boldsymbol{W}_{in,2}^\star) = f^\star(\cdot; \{\boldsymbol{A}^{\star(d)}\}_{d=1}^{D}, \boldsymbol{W}_B^\star, \boldsymbol{W}_C^\star, \boldsymbol{W}_{\boldsymbol{\Delta},\uparrow}^\star, \boldsymbol{W}_{\boldsymbol{\Delta},\downarrow}^\star, \boldsymbol{W}_{in,1}^\star, \boldsymbol{W}_{in,2}^\star).$$

$$\tag{11}$$

*Proof of Lemma 1.* To prove (11), we substitute (10) into (11), simplify the expression, and show that the equality holds under the following conditions:

$$\boldsymbol{W}_C^\star\boldsymbol{W}_{\text{in},1}^\star = \boldsymbol{W}_C\boldsymbol{W}_{\text{in},1} \tag{12}$$

$$\boldsymbol{W}_{\boldsymbol{\Delta},\uparrow}^\star\boldsymbol{W}_{\text{in},1}^\star = \boldsymbol{W}_{\boldsymbol{\Delta},\uparrow}\boldsymbol{W}_{\text{in},1}$$

$$\boldsymbol{W}_B^\star\boldsymbol{W}_{\text{in},1}^\star = \boldsymbol{W}_B\boldsymbol{W}_{\text{in},1}.$$

Since $W_{\text{S6}} = \begin{bmatrix} W_B \\ W_C \\ W_{\Delta,\uparrow} \end{bmatrix}$, the three conditions (12) can be compactly written as

$$W_{\text{S6}}^{\star} W_{\text{in},1}^{\star} = W_{\text{S6}} W_{\text{in},1}. \tag{13}$$

We now show that for any $W_{\text{S6}}$, there exists a matrix $W_{\text{in},1}$ that satisfies (13). By applying Singular Value Decomposition (SVD) to $W_{\text{S6}}$, we obtain:

$$W_{\text{S6}} = U \begin{bmatrix} \Sigma & O_{(2H+R)\times(D-2H-R)} \end{bmatrix} V^{\top}, \tag{14}$$

where $U \in \mathbb{R}^{(2H+R)\times(2H+R)}$, $\Sigma \in \mathbb{R}^{(2H+R)\times(2H+R)}$, and $V \in \mathbb{R}^{D\times D}$. The diagonal elements of $\Sigma$ are in decreasing order. We let

$$W_{\text{in},1} = V \begin{bmatrix} \Sigma^{-1} U^{\top} W_{\text{S6}}^{\star} W_{\text{in},1}^{\star} \\ Q \end{bmatrix}, \tag{15}$$

where $Q \in \mathbb{R}^{(D-2H-R)\times D}$ is an arbitrary matrix to be determined later. Plugging (14) and (15) back to $W_{\text{S6}} W_{\text{in},1}$ and simplifying results in

$$\begin{aligned} & W_{\text{S6}} W_{\text{in},1} \\ &= U \begin{bmatrix} \Sigma & O_{(2H+R)\times(D-2H-R)} \end{bmatrix} V^{\top} V \begin{bmatrix} \Sigma^{-1} U^{\top} W_{\text{S6}}^{\star} W_{\text{in},1}^{\star} \\ Q \end{bmatrix} \qquad ((14)\ \&\ (15)) \\ &= W_{\text{S6}}^{\star} W_{\text{in},1}^{\star}, \qquad\qquad\qquad\qquad\qquad\qquad\qquad\qquad (\text{Simplifying}) \end{aligned}$$

which demonstrates that (13) is satisfied and completes the proof. □

**Empirical Validation.** To experimentally verify Lemma 1, we conduct a small-scale experiment. Specifically, we train Mamba 130M on three GLUE tasks—RTE, MRPC, and CoLA—for ten epoch under two settings: (1) training only the linear projection layer ($W_{\text{in}}$), and (2) training the S6 modules ($W_B$, $W_C$, $W_{\Delta,\uparrow}$). We experiment with various learning rates and, for each configuration, repeated the best-performing setting five times to ensure robustness. As shown in Fig. 3 (training loss) and Table 15 (validation metrics), our results confirm that optimizing only the linear projection layer is as expressive as training the S6 layers. In fact, in all cases, training only the linear projection not only matches, but even outperforms S6 layer training and converges more quickly.

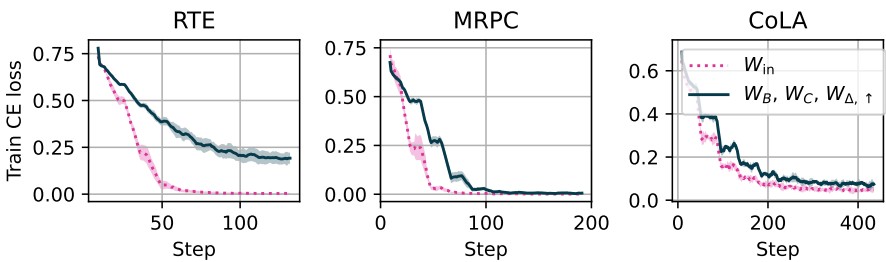

*Figure 3.* **Training cross-entropy loss, with the shaded area indicating the standard deviation, for fine-tuning linear projection layers versus S6 layers.** The results show that $W_{\text{in}}$ matches the expressivity of $W_B$, $W_C$, and $W_{\Delta,\uparrow}$, while also achieving faster convergence.

# D. Details of Sec. 5: SDT

## D.1. Understanding the Roles of State Matrix $A$, Input Transition Vector $B$, and Output Mapping Vector $C$ for a Single Channel in S4 Modules

**Problem Setting.** Inspired by Zeng & Lee (2024)'s theoretical analysis of LoRA's expressive power, we adopt a similar framework to explore the expressive potential of various parameters in the S4 model. Specifically, we assume a target model

| Layers | RTE | MRPC | CoLA |
|---|---|---|---|
| $W_{\text{in}}$ | 69.9 ± 1.2 | 82.4 ± 0.7 | 61.1 ± 2.3 |
| $W_B, W_C, W_{\Delta,\uparrow}$ | 68.3 ± 0.8 | 79.9 ± 1.2 | 54.9 ± 1.5 |

*Table 15.* **Mean and standard deviation of validation metrics for fine-tuning linear projection layers versus S6 layers.** The results demonstrate that $W_{\text{in}}$ effectively captures the expressivity of $W_B$, $W_C$, and $W_{\Delta,\uparrow}$, and achieves superior performance on validation metrics—accuracy for RTE and MRPC, and Matthews correlation coefficient for CoLA.

that performs well on the intended task and a frozen model, which may be either pretrained or randomly initialized. Our goal is to identify a parameter-efficient method to update the frozen model so that it becomes functionally equivalent to the target model. In alignment with Zeng & Lee (2024), we assume that the frozen model's capacity is equal to or exceeds that of the target model. This assumption is based on two main considerations: (i) analytical tractability, which necessitates that the frozen model must have the potential to match the functionality of the target model, and (ii) a practical rationale, given that the models typically used in practice are often overparameterized. Assume that both the target model and the frozen model are S4, with the target model having a hidden state dimension $H_\star$ and the frozen model having a hidden state dimension $H \geq H_\star$. Meanwhile, suppose that all the hidden dimensions of both models are valid, meaning that none of the parameter elements are zero. The target model, frozen model, and the updated model after tuning the parameters on the frozen model can be formulated using discretized parameters $\overline{A}, \overline{B}, C$ as follows:

$$\text{(Target model)} \quad f^\star(\boldsymbol{x})_n = \sum_{m=1}^{n} \boldsymbol{C}_\star \overline{\boldsymbol{A}}_\star^{m-n} \overline{\boldsymbol{B}}_\star x_m, \text{ where } \operatorname{diag}(\overline{\boldsymbol{A}}_\star), \overline{\boldsymbol{B}}_\star, \boldsymbol{C}_\star \in \mathbb{R}^{H_\star},$$

$$\text{(Frozen model)} \quad f_0(\boldsymbol{x})_n = \sum_{m=1}^{n} \boldsymbol{C} \overline{\boldsymbol{A}}^{m-n} \overline{\boldsymbol{B}} x_m, \text{ where } \operatorname{diag}(\overline{\boldsymbol{A}}), \overline{\boldsymbol{B}}, \boldsymbol{C} \in \mathbb{R}^{H},$$

$$\text{(Updated model)} \quad \hat{f}(\boldsymbol{x})_n = \sum_{m=1}^{n} \widehat{\boldsymbol{C}} \widehat{\overline{\boldsymbol{A}}}^{m-n} \widehat{\overline{\boldsymbol{B}}} x_m, \text{ where } \operatorname{diag}(\widehat{\overline{\boldsymbol{A}}}), \widehat{\overline{\boldsymbol{B}}}, \widehat{\boldsymbol{C}} \in \mathbb{R}^{H}.$$

**Parameter Efficiency Analysis on S4.** Let $\mathcal{P}^H$ denote the set of all $H \times H$ permutation matrices. Given this formulation, we present our first analysis of parameter efficiency for the S4 model in the following lemma. This analysis is based on the parameters after necessary discretization $(\overline{A}, \overline{B}, C)$. For the reader's convenience, we restate Lemma 2 below with minor notational changes to facilitate the proof.

**Lemma 2** (Minimal Parameter Adjustment for S4 Fine-Tuning). *Consider the parameters after discretization, i.e., $\overline{A}, \overline{B}, C$. To achieve functional equivalence between the updated model and the target model, i.e., $\hat{f} \equiv f^\star$, the minimum number of tunable parameters is:*

$$\min_{\boldsymbol{P} \in \mathcal{P}^H} \overbrace{\left\| \left[ \boldsymbol{P}^\top (\operatorname{diag}(\overline{\boldsymbol{A}}) \odot \overline{\boldsymbol{B}} \odot \boldsymbol{C}^\top) \right]_{(H_\star+1):H} \right\|_0}^{\text{eliminating redundant dimensions}} + \underbrace{\left\| \left[ \boldsymbol{P}^\top \overline{\boldsymbol{A}} \boldsymbol{P} \right]_{1:H_\star, 1:H_\star} - \overline{\boldsymbol{A}}_\star \right\|_0}_{\text{aligning the state matrix}} + \overbrace{\underbrace{\left\| \left[ \boldsymbol{P}^\top (\overline{\boldsymbol{B}} \odot \boldsymbol{C}^\top) \right]_{1:H_\star} - \overline{\boldsymbol{B}}_\star \odot \boldsymbol{C}_\star^\top \right\|_0}_{\text{aligning input-output interactions}}}^{\text{aligning used dimensions with target model}}.$$

*Proof of Lemma 2.* The key idea of this proof is straightforward. To facilitate the analysis and update the frozen model to be equivalent to the target model, we first equalize the number of hidden state dimensions between the two models. This is achieved by expanding the target model's $A_\star$, $B_\star$, and $C_\star$ to match the $H$ hidden state dimensions of the frozen model, padding the additional $H - H^\star$ dimensions with zeros.

Define $\odot$ as the element-wise product. We can express the target model as:

$$f^\star(\boldsymbol{x})_n = \sum_{m=1}^{n} \begin{bmatrix} \boldsymbol{C}_\star & \boldsymbol{0}^\top \end{bmatrix} \begin{bmatrix} \overline{\boldsymbol{A}}_\star & \boldsymbol{O} \\ \boldsymbol{O} & \boldsymbol{O} \end{bmatrix}^{n-m} \begin{bmatrix} \overline{\boldsymbol{B}}_\star \\ \boldsymbol{0} \end{bmatrix} x_m$$

$$= \sum_{m=1}^{n} \operatorname{diag}\left( \begin{bmatrix} \overline{\boldsymbol{A}}_\star & \boldsymbol{O} \\ \boldsymbol{O} & \boldsymbol{O} \end{bmatrix} \right)^{n-m} \left( \begin{bmatrix} \boldsymbol{C}_\star^\top \\ \boldsymbol{0} \end{bmatrix} \odot \begin{bmatrix} \overline{\boldsymbol{B}}_\star \\ \boldsymbol{0} \end{bmatrix} \right) x_m.$$

Consider any permutation matrix $\boldsymbol{P} \in \mathcal{P}^H$. Applying $\boldsymbol{P}$ to permute the frozen model leaves the model functionally unchanged:

$$f_0(\boldsymbol{x})_n = \sum_{m=1}^{n} \boldsymbol{C}\overline{\boldsymbol{A}}^{n-m}\overline{\boldsymbol{B}}x_m = \sum_{m=1}^{n} \boldsymbol{C}\boldsymbol{P}\left(\boldsymbol{P}^\top\overline{\boldsymbol{A}}\boldsymbol{P}\right)^{n-m}\boldsymbol{P}^\top\overline{\boldsymbol{B}}x_m$$

$$= \sum_{m=1}^{n} \operatorname{diag}\left(\boldsymbol{P}^\top\overline{\boldsymbol{A}}\boldsymbol{P}\right)^{n-m}\left(\left(\boldsymbol{P}^\top\boldsymbol{C}^\top\right) \odot \left(\boldsymbol{P}^\top\overline{\boldsymbol{B}}\right)\right)x_m.$$

Due to the convolution structure of $\overline{\boldsymbol{A}}$, two models are functionally equivalent if and only if $\boldsymbol{P}^\top\overline{\boldsymbol{A}}\boldsymbol{P}$ aligns with $\begin{bmatrix} \boldsymbol{A}_\star & \boldsymbol{O} \\ \boldsymbol{O} & \boldsymbol{O} \end{bmatrix}$, and $(\boldsymbol{P}^\top\boldsymbol{C}^\top) \odot (\boldsymbol{P}^\top\overline{\boldsymbol{B}})$ align with $\begin{bmatrix} \boldsymbol{C}_\star^\top \\ \boldsymbol{0} \end{bmatrix} \odot \begin{bmatrix} \boldsymbol{B}_\star \\ \boldsymbol{0} \end{bmatrix}$ for some $P \in \mathcal{P}^H$. If they are already matching or partially matched for certain entries, no updates are required for those entries; only the unmatched entries need to be updated. Then, the required trainable parameters for this permutation matrix $\boldsymbol{P}$ are:

$$\left\|\left[\boldsymbol{P}^\top(\operatorname{diag}(\overline{\boldsymbol{A}}) \odot \overline{\boldsymbol{B}} \odot \boldsymbol{C}^\top)\right]_{(H_\star+1):H}\right\|_0 + \left\|\left[\boldsymbol{P}^\top\overline{\boldsymbol{A}}\boldsymbol{P}\right]_{1:H_\star,1:H_\star} - \overline{\boldsymbol{A}}_\star\right\|_0 + \left\|\left[\boldsymbol{P}^\top(\overline{\boldsymbol{B}} \odot \boldsymbol{C}^\top)\right]_{1:H_\star} - \overline{\boldsymbol{B}}_\star \odot \boldsymbol{C}_\star^\top\right\|_0.$$

Optimizing the permutation matrix $\boldsymbol{P} \in \mathcal{P}^H$ yields the desired results. $\qquad\square$

This lemma highlights the significance of identifying essential hidden state dimensions. The term $\left\|\left[\boldsymbol{P}^\top(\operatorname{diag}(\overline{\boldsymbol{A}}) \odot \overline{\boldsymbol{B}} \odot \boldsymbol{C}^\top)\right]_{(H_\star+1):H}\right\|_0$ underscores the importance of excluding redundant dimensions. This can be achieved by either directly removing these dimensions from the state matrix $\overline{\boldsymbol{A}}$, or by updating $\overline{\boldsymbol{B}}$ or $\boldsymbol{C}$ to ensure that only the selected hidden state dimensions are utilized during the input transition or output mapping phases. Once redundant dimensions are filtered out, tuning only the essential dimensions is sufficient to align the updated model with the target model.

Furthermore, based on the lemma, the roles of the input transition vector $\overline{\boldsymbol{B}}$ and $\boldsymbol{C}^\top$ are nearly identical, as they consistently appear together as the combined term $\overline{\boldsymbol{B}} \odot \boldsymbol{C}^\top$, which is also discussed in Gupta et al. (2022). Consequently, one could opt to tune either $\overline{\boldsymbol{B}}$ or $\boldsymbol{C}$ exclusively or alternatively, split the indices into two groups, tuning $\overline{\boldsymbol{B}}$ for the first group and $\boldsymbol{C}$ for the second. Both vectors indicate how information from different hidden state dimensions is integrated, whereas $\overline{\boldsymbol{A}}$ plays a distinct role, determining how the hidden states are stored.

In practice, instead of directly using the discretized parameters $\overline{\boldsymbol{A}}, \overline{\boldsymbol{B}}, \boldsymbol{C}$, S4 is implemented using the continuous parameters $\boldsymbol{A}, \boldsymbol{B}, \boldsymbol{C}$ with step size $\Delta$. To provide further practical guidance on parameter tuning, the following two lemmas analyze the parameter efficiency of continuous parameters under different discretization methods. Two exemplary methods of discretization are bilinear and zero-order hold (ZOH):

$$\text{(Bilinear)} \begin{cases} \overline{\boldsymbol{A}} = (\boldsymbol{I} - \Delta/2\boldsymbol{A})^{-1}(\boldsymbol{I} + \Delta/2\boldsymbol{A}) \\ \overline{\boldsymbol{B}} = (\boldsymbol{I} - \Delta/2\boldsymbol{A})^{-1} \cdot \Delta\boldsymbol{B}, \end{cases} \quad \text{(ZOH)} \begin{cases} \overline{\boldsymbol{A}} = \exp(\Delta\boldsymbol{A}) \\ \overline{\boldsymbol{B}} = (\Delta\boldsymbol{A})^{-1}(\exp(\Delta\boldsymbol{A}) - \boldsymbol{I}) \cdot \Delta\boldsymbol{B}. \end{cases} \tag{16}$$

**Lemma 3** (Essential Continuous Parameter Set for S4 with Bilinear Discritization)**.** *Consider the parameters before discretization, i.e.,* $\boldsymbol{A}, \boldsymbol{B}, \boldsymbol{C}$, *which are subsequently discretized using bilinear discretization. To achieve functional equivalence between the updated model and the target model, i.e.,* $\hat{f} \equiv f^\star$, *it is sufficient to tune the following number of parameters:*

$$\min\nolimits_{\boldsymbol{P}\in\mathcal{P}^H} \overbrace{\left\|\left[\Delta\boldsymbol{P}^\top(\operatorname{diag}(\boldsymbol{I} + \Delta/2\boldsymbol{A}) \odot \boldsymbol{B} \odot \boldsymbol{C}^\top)\right]_{(H_\star+1):H}\right\|_0}^{\text{eliminating redundant dimensions}} + \overbrace{\underbrace{\left\|\left[\boldsymbol{P}^\top\boldsymbol{A}\boldsymbol{P}\right]_{1:H_\star,1:H_\star} - \boldsymbol{A}_\star\right\|_0}_{\text{aligning the state matrix}} + \underbrace{\left\|\left[\boldsymbol{P}^\top(\boldsymbol{B} \odot \boldsymbol{C}^\top)\right]_{1:H_\star} - \boldsymbol{B}_\star \odot \boldsymbol{C}_\star^\top\right\|_0}_{\text{aligning input-output interactions}}}^{\text{aligning used dimensions with target model}}.$$

*Proof of Lemma 3.* Combining Lemma 2 and the Bilinear discretization method in (16) yields the desired results. $\qquad\square$

**Lemma 4** (Essential Continuous Parameter Set for S4 with ZOH Discritization)**.** *Consider the parameters before discretization, i.e.,* $\boldsymbol{A}, \boldsymbol{B}, \boldsymbol{C}$, *which are subsequently discretized using ZOH discretization. To achieve functional equivalence between*

*the updated model and the target model, i.e., $\hat{f} \equiv f^\star$, it is sufficient to tune the following number of parameters:*

$$\min_{\boldsymbol{P} \in \mathcal{P}^H} \overbrace{\left\| [\Delta \boldsymbol{P}^\top (\text{diag}(\exp(\Delta \boldsymbol{A}) - \boldsymbol{I}) \odot \boldsymbol{B} \odot \boldsymbol{C}^\top)]_{(H_\star+1):H} \right\|_0}^{\text{eliminating redundant dimensions}} + \overbrace{\underbrace{\left\| [\boldsymbol{P}^\top \boldsymbol{A} \boldsymbol{P}]_{1:H_\star, 1:H_\star} - \boldsymbol{A}_\star \right\|_0}_{\text{aligning the state matrix}} + \underbrace{\left\| [\boldsymbol{P}^\top (\boldsymbol{B} \odot \boldsymbol{C}^\top)]_{1:H_\star} - \boldsymbol{B}_\star \odot \boldsymbol{C}_\star^\top \right\|_0}_{\text{aligning input-output interactions}}}^{\text{aligning used dimensions with target model}}.$$

*Proof of Lemma 4.* Combining Lemma 2 and the ZOH discretization method in (16) yields the desired results. □

The insights provided by Lemma 3 and Lemma 4 are the same as those provided by Lemma 2. The analysis here supports the second step of SDT-P presented in Sec. 5.

### D.2. Extension to Deep S4 Models

Our previous analysis focused on single-channel S4 models. We now expand our investigation to more complex scenarios involving deep S4 models for both target and frozen architectures, incorporating $D$ channels and varying layer depths. In this section, in addition to SDT-P, we introduce SDT+. The key distinction between SDT+ and SDT-P lies in their treatment of linear projection matrices. SDT-P only operate on SSM modules and additionally requires applying LoRA to modify the linear projection matrices. In contrast, SDT+ applies SDT-P on SSM modules while also updates the columns of weight matrices corresponding to the updatable channels identified through Alg. 2. It is worth noting that the linear projection matrix updates in SDT+ are inherently low-rank, making it a specialized case of SDT-P combined with LoRA. Our analysis starts with SDT+, and it automatically applies to SDT-P combined with LoRA.

In this analysis, we assume that each input token $x_t$ belongs to $\mathcal{X}$, a bounded subset of $\mathbb{R}^D$, and that the length of the input sequence is finite. Let the frozen model have $L$ layers, and the target model have $L^\star$ layers, where $L \geq L^\star$. Similar to the technique used in Zeng & Lee (2024) and Giannou et al. (2023). The basic idea of updating the frozen model to match the functionality of the target model is to utilize every $\lceil L/L^\star \rceil$ layers of the frozen model to approximate every layer of the target model. We start introducing this proof idea from the simplest case where $L^\star = 1, L = D$. In this scenario, we can simply choose one different channel to tune and maintain all other channels at zero at every layer. The outputs from the various channels of the deep S4 layers are then combined through a residual connection. This proof idea inspires us to perform channel selection and make use of the residual connections, which is the first and third step of SDT-P presented in Sec. 5. Building on this idea, we present the following results for when the target model has only $L^\star = 1$ layer, and $L = D = 2$.

**Lemma 5.** *Consider a $D$-dimensional input sequence. Assume that the linear layers in the model have linear activation functions. Using SDT+, any deep S4 model with $H$ hidden states per channel and $L$ layers can be updated to accurately present any target one-layer deep S4 model without residual connections, having a reduced hidden state dimension $H^\star < H$. Then this can be achieved by selectively fine-tuning at most $\lceil D/L \rceil$ channels, $H^\star$ hidden states, and residual connections at each layer, while additionally fully fine-tuning the linear projection matrix of the last layer only.*

*Proof of Lemma 5.* In this proof, we start by considering the case where $L = D$. In this case, we update a single distinct channel for each layer while setting the other channels to zero. Essentially, we modify the frozen model so that each layer corresponds to and functions as an individual channel in the target model. To be more specific, we fully update the first channel in the first layer to match the first channel of the target model, second channel in the second layer to match the second channel of the target model, so on and so forth.

For the $l$-th layer of the frozen model , we append subscript $l$ to all parameters of the deep S4 layer as introduced in (4). For the $d$-th channel, corresponding notations are denoted with a superscript $(d)$. We define the $t$-th intermediate output token of the $l$-th deep S4 layer as $\boldsymbol{z}_{l,t} \in \mathbb{R}^D$. Additionally, the updated S4 module in layer $l$ is denoted as $\widehat{\text{S4}}_l$, with $\widehat{\text{S4}}_{l,t}$ referring specifically to the sub-function that outputs the $t$-th token. Therefore, for the $t$-th intermediate output token of the $l$-th deep S4

S4 layer of the updated model can be written as

$$
\begin{aligned}
\boldsymbol{z}_{l,t} &= \widehat{\boldsymbol{W}}_l \cdot \widehat{\text{S4}}_{l,t}(\boldsymbol{z}_{l-1,1}, \ldots, \boldsymbol{z}_{l-1,t}) + \widehat{\boldsymbol{\beta}}_l + \widehat{\boldsymbol{u}}_l \odot \boldsymbol{z}_{l-1,t} \\
&= \widehat{\boldsymbol{W}}_l \cdot
\begin{bmatrix}
\widehat{\text{S4}}_{l,t}^{(1)}(z_{l-1,1}^{(1)}, \ldots, z_{l-1,t}^{(1)}) \\
\vdots \\
\widehat{\text{S4}}_{l,t}^{(D)}(z_{l-1,1}^{(D)}, \ldots, z_{l-1,t}^{(D)})
\end{bmatrix}
+ \widehat{\boldsymbol{\beta}}_l + \widehat{\boldsymbol{u}}_l \odot \boldsymbol{z}_{l-1,t},
\end{aligned}
$$

where $\widehat{\boldsymbol{W}}_l \in \mathbb{R}^{D \times D}, \widehat{\boldsymbol{\beta}}_l \in \mathbb{R}^D$ are the updated weight and biases of the $l$-th layer of the frozen model, and $\widehat{\boldsymbol{u}}_l \in \mathbb{R}^D$ is the updated residual connection weight of the frozen model.

**For layers** $l < L = D$. We follow the steps provided in Sec. 5 to update the $l$-th layer of the frozen model such that it functionally equivalent to the $l$-th channel of the target model. For the reader's convinence, we restate our strategies here:

- **(Channel Selection)** Select $D' \leq D$ ($D' = 1$ here) important channels for making predictions. Any channel $d$ that is not utilized will have their corresponding $\boldsymbol{C}^{(d)}$ set to zero, eliminating the need to update parameters for $\boldsymbol{A}^{(d)}$ and the $d$-th column of $\boldsymbol{W}$. To be more specific, we let $\boldsymbol{C}^{(d)} = \boldsymbol{0}$ for all $d \neq l$ in this scenario.

- **(Hidden State Selection)** Within the selected channels, select $H' \leq H$ important hidden states. For any hidden state that is not used within a selected channel $d$, the corresponding element in $\boldsymbol{C}^{(d)}$ will be set to zero, thus eliminating the need to tune the corresponding element in $\boldsymbol{A}^{(d)}$. To be more specific, we can achieve $\widehat{\text{S4}}_{l,t}^{(l)}(\cdot) = \text{S4}_{\star,t}^{(l)}(\cdot)$ by Lemma 2.

- **(Residual and Bias Tuning)** Regardless of other selections, SDT consistently tunes the coefficients of residual connections and biases in linear projections, as these components contain a negligible number of parameters. In this scenario, we let $\widehat{\boldsymbol{\beta}}_l = \boldsymbol{0}$, $\widehat{\boldsymbol{u}}_l = \begin{bmatrix} \underbrace{1 \quad \cdots \quad 1}_{l-1 \text{ elements}} & 0 & \underbrace{1 \quad \cdots \quad 1}_{D-l \text{ elements}} \end{bmatrix}^{\top}$.

This construction yields

$$
\boldsymbol{z}_{l,t} = \begin{bmatrix} z_{l-1,t}^{(1)} & \cdots & z_{l-1,t}^{(l-1)} & \text{S4}_{\star,t}^{(l)}(z_{l,1}^{(l)}, \ldots, z_{l,t}^{(l)}) & z_{l-1,t}^{(l+1)} & \cdots & z_{l-1,t}^{(D)} \end{bmatrix}^{\top}.
$$

Consequently, only the $l$-th channel is active in the $l$-th layer, while all other layers function as identity mappings, propagating the output of the preceding layer without modification.

**For layer** $l = L = D$. Based on the setup of the first $L - 1$ layers, we have

$$
\boldsymbol{z}_{L-1,t} = \begin{bmatrix} \text{S4}_{\star,t}^{(1)}(x^{(1)}) & \cdots & \text{S4}_{\star,t}^{(L-1)}(x^{(L-1)}) & x^{(L)} \end{bmatrix}^{\top}.
$$

For the last layer, we let

$$
\widehat{\boldsymbol{W}}_L = \boldsymbol{W}_\star, \quad \widehat{\boldsymbol{\beta}}_L = \boldsymbol{\beta}_\star, \quad \widehat{\boldsymbol{u}}_L = \boldsymbol{0},
$$
$$
\widehat{\text{S4}}_{L,t}^{(L)}(\cdot) = \text{S4}_{\star,t}^{(L)}(\cdot), \text{ which can be achieved by Lemma 2.}
$$

It is easy to verify that the output of the updated frozen model is identical to the output of the target model, i.e.,

$$
\boldsymbol{y}_t = \boldsymbol{z}_{L,t} = \boldsymbol{W}_\star \begin{bmatrix} \text{S4}_{\star,t}^{(1)}(x^{(1)}) & \cdots & \text{S4}_{\star,t}^{(L-1)}(x^{(L-1)}) & \text{S4}_{\star,t}^{(L)}(x^{(L)}) \end{bmatrix}^{\top} + \boldsymbol{\beta}_\star.
$$

Thus far, we have demonstrated that the statement holds when $L = D$. This analysis can be readily extended to cases where $L \neq D$ by tuning $\lceil D/L \rceil$ channels at each layer. For example, when $L = D/2$, we can tune two channels per layer using a construction similar to the one described above. This generalization completes the proof. $\square$

**Theorem 2** (Expressive Power of SDT+ on Deep S4 Models). *Consider a $D$-dimensional input sequence. Assume that the linear layers in the model have linear activation functions. Using SDT+, any deep S4 model with $H$ hidden states per channel and $L$ layers can be updated to accurately present any target deep S4 model without residual connections, having a reduced hidden state dimension $H^\star < H$, and fewer layers $L^\star < L$. This can be achieved by selectively fine-tuning at most $\lceil DL^\star/L \rceil$ channels, $H^\star$ hidden states, and residual connections at each layer.*

*Proof of Theorem 2.* We update every $\lceil D/L \rceil$ layers of the frozen model to approximate each layer of the target model. By applying Lemma 5 iteratively to each set of $\lceil D/L \rceil$ layers, we obtain the desired result. $\square$

Theorem 2 leads to the following result, which represents the deep S4 model case of Theorem 1.

**Theorem 3** (Expressive Power of SDT-P on Deep S4 Models). *Consider a $D$-dimensional input sequence. Assume that the linear layers in the model have linear activation functions. Using SDT-P, any deep S4 model with $H$ hidden states per channel and $L$ layers can be updated to accurately present any target deep S4 model without residual connections, having a reduced hidden state dimension $H^\star < H$, and fewer layers $L^\star < L$. This can be achieved by selectively fine-tuning at most $\lceil DL^\star/L \rceil$ channels, $H^\star$ hidden states on SSM modules, applying rank-$\lceil \frac{L}{L^\star} \rceil$ updates on linear projection matrices and updating residual connections and biases at each layer, while additionally fully fine-tuning the linear projection matrix of the last layer only.*

*Proof of Theorem 1.* Since SDT+ is a special case of SDT-P, Theorem 2 directly implies the desired statement. $\square$

### D.3. Extension to S6

In this section, we extend the discussion of SDT-P and SDT+ to S6, following the same logic. We begin by proving results for SDT+ in the scenario where the target model consists of only a single layer. In doing so, we extend Theorem 3 to apply to deep S6 models by first generalizing Lemma 5 to Lemma 6.

**Lemma 6.** *Consider a $D$-dimensional input sequence. Assume that the linear layers in the model have linear activation functions. Using SDT+, any deep S6 model with $H$ hidden states per channel and $L$ layers can be updated to accurately present any target one-layer deep S6 model without residual connections, having a reduced hidden state dimension $H^\star < H$. Then this can be achieved by selectively fine-tuning at most $\lceil D/L \rceil$ channels, $H^\star$ hidden states, and residual connections at each layer, while additionally fully fine-tuning the linear projection matrix of the last layer only.*

*Proof of Lemma 6.* To demonstrate this, we can follow the same proof strategy as in the proof of Lemma 5. In particular, the $t$-th intermediate output token of the $l$-th deep S6 layer of the updated model can be similarly written as

$$\boldsymbol{z}_{l,t} = \widehat{\boldsymbol{W}}_l \cdot \widehat{\text{S6}}_{l,t}(\boldsymbol{z}_{l-1,1}, \ldots, \boldsymbol{z}_{l-1,t}) + \widehat{\boldsymbol{\beta}}_l + \widehat{\boldsymbol{u}}_l \odot \boldsymbol{z}_{l-1,t}$$

$$= \widehat{\boldsymbol{W}}_l \cdot \begin{bmatrix} \widehat{\text{S6}}_{l,t}^{(1)}(z_{l-1,1}^{(1)}, \ldots, z_{l-1,t}^{(1)}) \\ \vdots \\ \widehat{\text{S6}}_{l,t}^{(D)}(z_{l-1,1}^{(D)}, \ldots, z_{l-1,t}^{(D)}) \end{bmatrix} + \widehat{\boldsymbol{\beta}}_l + \widehat{\boldsymbol{u}}_l \odot \boldsymbol{z}_{l-1,t},$$

where $\widehat{\boldsymbol{W}}_l \in \mathbb{R}^{D \times D}, \widehat{\boldsymbol{\beta}}_l \in \mathbb{R}^D$ are the updated weight and biases of the $l$-th layer of the frozen model, and $\widehat{\boldsymbol{u}}_l \in \mathbb{R}^D$ is the updated residual connection weight of the frozen model.

**For layers $l < L = D$.** We follow the steps provided in Sec. 5 to update the $l$-th layer of the frozen model such that it functionally equivalent to the $l$-th channel of the target model. For the reader's convinence, we restate our strategies here:

- **(Channel Selection)** Select $D' \le D$ ($D' = 1$ here) important channels for making predictions. For any channel $d$ that is not utilized, rather than directly setting the corresponding $\boldsymbol{C}^{(d)}$ to zero as in the deep S4 model, we instead set $\boldsymbol{\beta}_{\boldsymbol{\Delta}}^{(d)}$ to be sufficiently large. According to the computation of SSM parameters described in (17), this ensures that $\overline{\boldsymbol{B}}_n^{(d)}$ is set to zero for all $d \neq l$ in this scenario. This approach is equivalent to setting $\boldsymbol{C}^{(d)}$ to zero, as both result in the channel producing all zeros.

- **(Hidden State Selection)** Within the selected channels, select $H' \leq H$ important hidden states. For any hidden state that is not used within a selected channel $d$, the corresponding entries in $\boldsymbol{A}^{(d)}$ will be set to sufficiently small. To be more specific, we can achieve $\widehat{\text{S6}}_{l,t}^{(l)}(\cdot) = \text{S6}_{\star,t}^{(l)}(\cdot)$ by leveraging the discretized parameters. Lemma 2 provides the conditions for this equality to hold by updating $\overline{\boldsymbol{A}}$, $\overline{\boldsymbol{B}}$, and $\boldsymbol{C}$ for the $l$-th channel, where these parameters are computed as follows:

$$\boldsymbol{\Delta}_n = \text{softplus}(\boldsymbol{W}_{\boldsymbol{\Delta},\downarrow}\boldsymbol{W}_{\boldsymbol{\Delta},\uparrow}\boldsymbol{x}_n + \boldsymbol{\beta}_{\boldsymbol{\Delta}}), \tag{17}$$
$$\overline{\boldsymbol{A}}_n^{(d)} = \exp(\Delta_n^{(d)}\boldsymbol{A}^{(d)}), \quad \overline{\boldsymbol{B}}_n^{(d)} = \Delta_n^{(d)}\boldsymbol{W}_B\boldsymbol{x}_n, \quad \boldsymbol{C}_n = \boldsymbol{W}_C\boldsymbol{x}_n,$$

Therefore, we can achieve $\widehat{\text{S6}}_{l,t}^{(l)}(\cdot) = \text{S6}_{\star,t}^{(l)}(\cdot)$ by only updating the corresponding values or columns of the weight matrices for each channel and dimension.

- **(Residual and Bias Tuning)** Regardless of other selections, SDT+ consistently tunes the coefficients of residual connections and biases in linear projections, as these components contain a negligible number of parameters. In this scenario, we let $\widehat{\boldsymbol{\beta}}_l = \boldsymbol{0}$, $\widehat{\boldsymbol{u}}_l = \begin{bmatrix} \underbrace{1 \quad \cdots \quad 1}_{l-1 \text{ elements}} & 0 & \underbrace{1 \quad \cdots \quad 1}_{D-l \text{ elements}} \end{bmatrix}^{\top}$.

This construction yields

$$\boldsymbol{z}_{l,t} = \begin{bmatrix} z_{l-1,t}^{(1)} & \cdots & z_{l-1,t}^{(l-1)} & \text{S4}_{\star,t}^{(l)}(z_{l,1}^{(l)},\ldots,z_{l,t}^{(l)}) & z_{l-1,t}^{(l+1)} & \cdots & z_{l-1,t}^{(D)} \end{bmatrix}^{\top}.$$

For the remaining layers, following the same steps leads to the desired results. $\qquad\square$

Therefore, we similarly obtain the following two results.

**Theorem 4** (Expressive Power of SDT+ on Deep S6 Models)**.** *Consider a $D$-dimensional input sequence. Assume that the linear layers in the model have linear activation functions. Using SDT+, any deep S6 model with $H$ hidden states per channel and $L$ layers can be updated to accurately present any target deep S6 model without residual connections, having a reduced hidden state dimension $H^{\star} < H$, and fewer layers $L^{\star} < L$. This can be achieved by selectively fine-tuning at most $\lceil DL^{\star}/L \rceil$ channels, $H^{\star}$ hidden states, and residual connections at each layer.*

**Theorem 5** (Expressive Power of SDT-P on Deep S6 Models)**.** *Consider a $D$-dimensional input sequence. Assume that the linear layers in the model have linear activation functions. Using SDT-P, any deep S6 model with $H$ hidden states per channel and $L$ layers can be updated to accurately present any target deep S6 model without residual connections, having a reduced hidden state dimension $H^{\star} < H$, and fewer layers $L^{\star} < L$. This can be achieved by selectively fine-tuning at most $\lceil DL^{\star}/L \rceil$ channels, $H^{\star}$ hidden states on SSM modules, applying rank-$\lceil \frac{L}{L^{\star}} \rceil$ updates on linear projection matrices and updating residual connections and biases at each layer, while additionally fully fine-tuning the linear projection matrix of the last layer only.*

Combining Theorem 3 and 5 leads to Theorem 1.

### D.4. Sparse Dimension Tuning and Pruning (SDT-P)

Algorithm 2 is our extended algorithm, which includes setting dimensions to zero. However, in practical settings, setting channels to zero is not necessary and omitting it reduces number of hyperparameters, as pruning parameters is effectively equivalent to training them to zero.

---

**Algorithm 2** Dimension Selection Algorithm of SDT-P

---

**Input:** A small subset of dataset $\mathcal{D}$, warmup epochs $E$, number of layers $L$, total channels $D$, total states $H$, state sparsity $\beta_0$, channel sparsity $\alpha_0$, state update fraction $\beta$, channel update fraction $\alpha$

```
/* Warmup Epochs                                                                      */
```
Perform full update on SSM modules using $\mathcal{D}$ for $E$ epochs `/* Categorize dimensions      */`
**for** $l = 1$ **to** $L$ **do**

    `/* Set dimensions as zero                                                          */`

    Sort channels based on $\|\overline{\boldsymbol{A}}^{(d)}\|$

    Select final $(1 - \beta_0)D$ channels as zero channels and denote non-zero channels as set $\mathbb{D}$

    **for** $d \in \mathbb{D}$ **do**

        Sort states based on magnitude of $\bar{A}_h^{(d)}$ at each state dimension

        Select final $(1 - \alpha_0)H$ states as zero states and denote non-zero states as set $\mathbb{H}$

    `/* Unfreeze dimensions                                                             */`

    Sort non-zero channels $\mathbb{D}$ based on changes of $\|\overline{\boldsymbol{A}}^{(d)}\|$

    Select the top $\beta|\mathbb{D}|$ channels as updatable, denoted by $\mathbb{D}'$

    **for** $d \in \mathbb{D}'$ **do**

        Sort non-zero state dimensions based on changes of $\|\overline{\boldsymbol{A}}^{(d)}\|$

        Select the top $\alpha|\mathbb{H}|$ states as updatable at the $d$-th channel

---

### D.5. Extension to S5

In this part, we extend Lemma 2 and Theorem 1 to two corresponding results for S5. The extension follows the same procedure as the previous proof, so we omit the details here.

**Lemma 7** (Minimal Parameter Adjustment for S5 Fine-Tuning). *Assume all hidden dimensions of the target model $f^\star$ are non-zero, i.e., all elements of $\mathrm{diag}(\overline{\boldsymbol{A}}_\star) \odot \overline{\boldsymbol{B}}_\star^{(d)} \odot \boldsymbol{C}_\star^{(d)}$ are non-zero. To update frozen model $f_0$ such that it becomes functionally equivalent to the target model $f^\star$, the minimum number of tunable parameters is:*

$$\min_{\overline{\boldsymbol{A}}, \overline{\boldsymbol{B}}, \boldsymbol{C}} \left\| \left[ \overline{\boldsymbol{A}} \right]_{1:H_\star, 1:H_\star} - \overline{\boldsymbol{A}}_\star \right\|_0 + \sum_{d=1}^{D} \left( \overbrace{\left\| \left[ \mathrm{diag}(\overline{\boldsymbol{A}}) \odot \overline{\boldsymbol{B}}^{(d)} \odot \boldsymbol{C}^{(d)\top} \right]_{(H_\star+1):H} \right\|_0}^{\text{eliminating redundant dimensions}} + \overbrace{\left\| \left[ \overline{\boldsymbol{B}}^{(d)} \odot \boldsymbol{C}^{(d)\top} \right]_{1:H_\star} - \overline{\boldsymbol{B}}_\star^{(d)} \odot \boldsymbol{C}_\star^{(d)\top} \right\|_0}^{\text{aligning remaining dimensions with target model}} \right),$$

$$\text{subject to} \quad (\overline{\boldsymbol{A}}, \overline{\boldsymbol{B}}, \boldsymbol{C}) \in \{ (\boldsymbol{P}^\top \overline{\boldsymbol{A}}_0 \boldsymbol{P}, \boldsymbol{P}^\top \overline{\boldsymbol{B}}_0, \boldsymbol{C}_0 \boldsymbol{P}) : \boldsymbol{P} \text{ is a permutation matrix} \}.$$

**Theorem 6** (Expressive Power of SDT-P with LoRA on Simplified SSM-based Models). *Assume all layers use linear activations. Let $f_0$ be a frozen deep S4 S5, or S6 model with $L$ layers, each containing $H$ hidden states per channel. Let $f^\star$ be a smaller target model of the same type (S4, S5 or S6), with no residual connections, $L^\star < L$ layers, and $H^\star < H$ hidden states per channel. Then, there exists a set of parameter updates to $f_0$ satisfying the following conditions such that for any finite-length input sequence $\boldsymbol{X} = (\boldsymbol{x}_1, \ldots, \boldsymbol{x}_N)$ with $\boldsymbol{x}_n \in \mathcal{X} \subset \mathbb{R}^D$, where $\mathcal{X}$ is bounded, the resulting model $f$ satisfies $f(\boldsymbol{X}) = f^\star(\boldsymbol{X})$:*

1. *(**SDT-P on SSM**) In each SSM module, update at most $\lceil DL^\star/L \rceil$ channels. Within each updated channel, fine-tune at most $H^\star$ hidden states and set the rest to zero.*
2. *(**LoRA$^\star$ on Linear Projections**) Apply rank-$\lceil L/L^\star \rceil$ updates to each linear projection matrix.*
3. *(**Minimal Additional Updates**) Modify only the residual connections, per-layer biases, and the final-layer output projection.*

### D.6. Memory Usage and Runtime Analysis of SDT

**Memory Usage Analysis.** To assess the memory usage of SDT and LoRA, we conducted experiments on four different models, including both SSM and hybrid architectures. For each model and method, a dataset was generated with 2,500 batches of data samples, each batch comprising a random sequence of 1,500 tokens. The simulation was repeated four times, including dataset generation. All experiments were carried out on a single H100 GPU, and the reported metrics represent

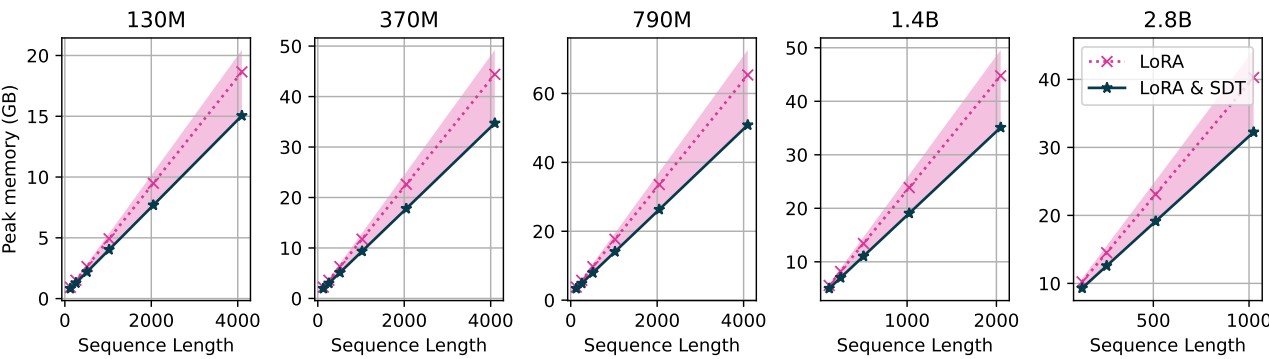

*Figure 4.* **Peak memory usage during training as a function of context length for different Mamba model sizes.** Each line represents the mean across three configurations; shaded regions indicate min–max ranges. SDT is consistently more memory-efficient than LoRA when applying to SSM module.

averages across the four simulations. Consistent with our previous experiments, we used the original hyperparameter settings, ensuring that SDT includes similar trainable parameters than LoRA. The memory usage of LoRA and SDT is presented in Table 16. Our observations indicate that SDT requires less memory than LoRA. This difference can be attributed to the design of the LoRA adapters, which involve matrix multiplication of two low-rank matrices. In contrast, tuning SSM with the same number of parameters does not require any matrix multiplication, resulting in lower memory usage.

| Memory Usage (GB) | Mamba-130M | Mamba-1.4B | Jamba-Tiny-319M | Jamba-Mini-52B |
|---|---|---|---|---|
| LoRA | 7.753 | 37.167 | 7.207 | 71.986 |
| LoRA & SDT | **5.738** | **26.491** | **6.605** | **67.193** |

*Table 16.* **Memory usage comparison between SDT and LoRA on various models. Bold** numbers indicate the lowest memory usage for each model.

To provide a more fine-grained view, we further analyze how sequence length affects peak memory usage for different Mamba model sizes, as shown in Fig. 4. We measure the memory required to process a single training batch with varying context lengths using randomly generated data. Each batch contains four examples, with 90% of tokens used as input and 10% as output (loss is computed only on the output tokens). The experimental settings for both LoRA and SDT follow the setup described in Section 6.2. We evaluate three configurations for each method, matched in parameter budget. In the plot, each line represents the average across the three configurations, and the shaded region for LoRA shows the range (minimum to maximum). SDT shows negligible variance across configurations, so no shading is included. All models are trained for 500 iterations, and results are averaged over these iterations. Experiments were conducted on an NVIDIA H100 80GB GPU.

**Runtime Analysis.** We similarly analyze the latency of LoRA and SDT, using the same experimental setup as in Table 16. Fine-tuning with SDT consists of two stages: (1) dimension selection and (2) standard training. In this study, we first compare the runtime of SDT and LoRA during stage 2 (training) and then evaluate the additional runtime introduced by SDT during stage 1 (dimension selection). Our results show that the dimension selection stage adds only marginal runtime overhead, and SDT is more efficient than LoRA in standard training.

*Training:* When the channels and states have been selected, the training of SDT is faster than LoRA when the same number of trainable parameters are considered.

The runtimes are reported in Table 17. We observe that, despite having similar trainable parameters, SDT is faster than LoRA. We attribute this to the fact that LoRA introduces additional FLOPs due to the extra matrix multiplication operations required for each update (specifically, the multiplication of two low-rank matrices).

*Dimension Selection:* For dimension selection, our method first performs Initial Subset Training, and then selects the dimensions based on the magnitude of parameter changes across different dimensions.

1. *Initial Subset Training:* We update the model by going through only a subset of the dataset (e.g., 3% of batches in DART

| Avg. Runtime (Seconds) | Mamba-130M | Mamba-1.4B | Jamba-Tiny-319M | Jamba-Mini-52B |
|---|---|---|---|---|
| LoRA | $410.0 \pm 80.0$ | $2060.0 \pm 135.0$ | $352.5 \pm 107.5$ | $3427.5 \pm 185.0$ |
| LoRA & SDT | $\mathbf{330.0} \pm 77.5$ | $\mathbf{1697.5} \pm 87.5$ | $\mathbf{257.5} \pm 72.5$ | $\mathbf{3065.0} \pm 232.5$ |

*Table 17.* **Runtime comparison of SDT and LoRA during stage 2 (training).**

experiments), which is sufficient in practice.

2. *Magnitude-Based Dimension Selection:* After the subset training, we select dimensions based on the magnitude of parameter changes observed.

In this experiment, we simulate a real scenario using a dataset with 2,500 batches, considering a small subset containing 125 batches (5% of the full dataset). We repeat the experiments 80 times, and the reported numbers are averaged across these simulations. Table 18 demonstrates that the dimension selection stage adds only negligible runtime.

| Avg. Runtime (Seconds) | Mamba-130M | Mamba-1.4B | Jamba-Tiny-319M | Jamba-Mini-52B |
|---|---|---|---|---|
| Initial Subset Training | $16.250 \pm 3.880$ | $85.250 \pm 5.130$ | $15.750 \pm 1.000$ | $163.630 \pm 10.120$ |
| Magnitude-Based Dimension Selection | $0.280 \pm 0.000$ | $0.520 \pm 0.120$ | $0.090 \pm 0.000$ | $0.240 \pm 0.040$ |
| Total Time | $16.530 \pm 3.880$ | $85.770 \pm 5.250$ | $15.840 \pm 1.000$ | $163.870 \pm 10.160$ |
| Proportion of Training 1 Epoch | $0.050\times$ | $0.051\times$ | $0.062\times$ | $0.053\times$ |
| Proportion of Training 5 Epoch | $\mathbf{0.010}\times$ | $\mathbf{0.010}\times$ | $\mathbf{0.012}\times$ | $\mathbf{0.011}\times$ |

*Table 18.* **Runtime comparison of SDT and LoRA during stage 1 (dimension selection).**

We further examine how runtime varies with sequence length, using the same experimental setup as in the memory analysis (Fig. 4). Our results in Fig. 5 show that SDT consistently outperforms LoRA in training speed when applied to SSM modules.

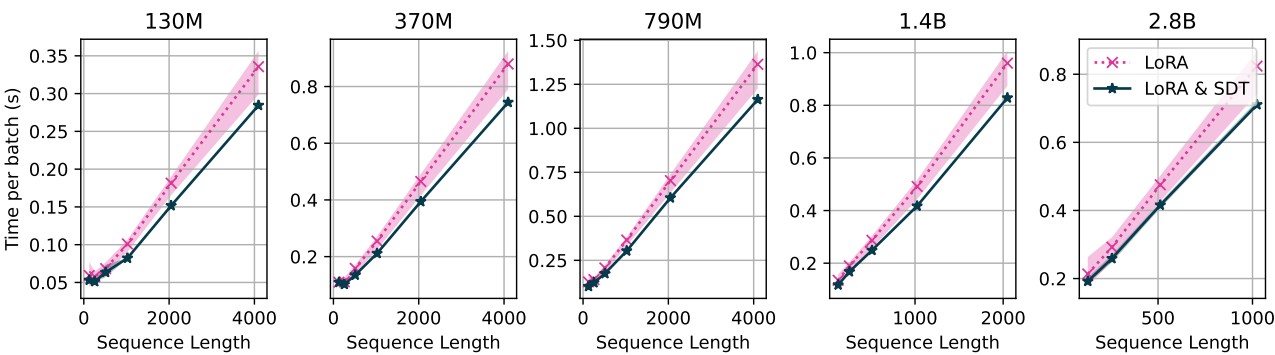

*Figure 5.* **Average training time per batch across different sequence lengths and Mamba model sizes.** Each line represents the mean across three configurations; shaded regions indicate min–max ranges. SDT is consistently faster than LoRA when applying to SSM module.

# E. Expanded Sec. 6: Evaluation of SDT

## E.1. Experiments on Deep S4 Models

**Synthetic.** For selecting channels and hidden states, we initiate with a warmup learning rate between $1e-2$ and $1e-3$ and conduct 20 warmup iterations. Learning rates are adjusted between $5e-2$, $1e-2$, $5e-3$, and $1e-3$. We apply LoRA with ranks of 2 and 4 to the SSM and with ranks of 4, 8, and 16 to the linear projection matrices. Non-zero states are selected from the sets $\{4, 8\}$, and non-zero channels from $\{8, 16\}$.

| Method | # Params (%) | Accuracy |
|---|---:|---|
| Frozen | 0.00 | 73.9 |
| LoRA (Proj) | 16.00 | 77.6 |
| LoRA (S4+Proj) | 15.52 | 77.6 |
| LoRA & SDT | 11.17 | **78.0** |
| Full Fine-Tuning | 100.00 | 77.6 |

*Table 19.* **Accuracy comparison between SDT and LoRA on deep S4 models for CIFAR-10 (Krizhevsky et al., 2009).**

In addition, we compare the convergence speed of LoRA and SDT in terms of training loss for sequence lengths in $\{100, 500, 1000\}$. We plot the MSE of both methods against wall-clock time. As shown in Fig. 6, SDT consistently converges to a lower loss faster than LoRA across all tested sequence lengths.

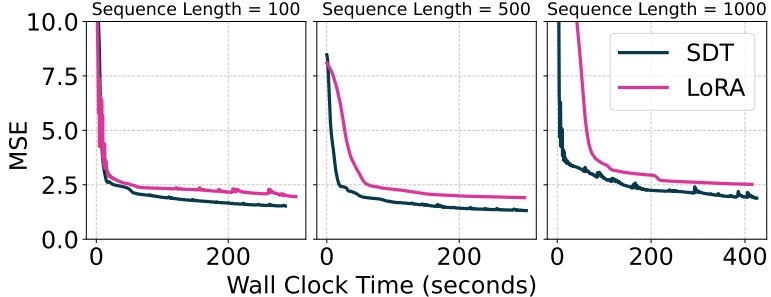

*Figure 6.* **Comparison of SDT and LoRA for tuning S4 in deep S4 models, where LoRA is applied to linear projection matrices.** Results are shown across varying sequence lengths under different time budgets in synthetic experiments.

**CIFAR-10 (Krizhevsky et al., 2009).** Previous work (Dinh et al., 2022) demonstrates that large language models can be fine-tuned for image classification tasks. In this study, we investigate the adaptation of SSMs for computer vision, focusing on experiments conducted with the CIFAR-10 dataset (Krizhevsky et al., 2009). We employ an eight-layer deep S4 model with a hidden state dimension of 16 and a model dimension of 64. Since pretrained deep S4 models are not available, we simulate a pretrained scenario by fully updating the model for 50 epochs first, then subsequently evaluating the PEFT methods over an additional 5 epochs. We adhere to the preprocessing steps for CIFAR-10 as outlined by Gu et al. (2022a). The LoRA ranks for linear projection matrices are tuned among $\{1, 2, 4, 8, 16\}$, and for the S4 component, ranks are set from $\{1, 2, 4\}$. Non-zero states are chosen from $\{8, 12, 16\}$, and non-zero channels from $\{48, 64\}$. The warmup phase includes 1 epoch with a learning rate of $1e - 2$. For linear projection matrices, LoRA ranks are explored at $\{2, 4, 8, 16\}$, and for the SSM, ranks at $\{2, 4, 8\}$. All state dimensions are updated, and channel dimensions considered for updates are $\{4, 8, 16, 32\}$. The results, as reported in Table 19, indicate that SDT outperforms LoRA with fewer trainable parameters.

### E.2. Experiments on Mamba-II, Jamba, and LoRA+

**Additional Experimental Details.** In this paragraph, we provide further experimental details. Unless otherwise stated, our experiment setting is identical to Sec. C.1. For LoRA, we consider three different LoRA configurations at each layer, targeting the primary parameters of Mamba. Specifically, we focus on the following matrices: $\boldsymbol{W}_{\text{out}}$ (output linear projection), $\boldsymbol{W}_B, \boldsymbol{W}_C$ (weight matrices for computing input-dependent $\boldsymbol{B}_n, \boldsymbol{C}_n$), and $\boldsymbol{W}_{\boldsymbol{\Delta}, \downarrow}, \boldsymbol{W}_{\boldsymbol{\Delta}, \uparrow}$ (down and up projection matrices of LoRA adapters for computing $\boldsymbol{\Delta}$). The three LoRA application methods are: (i) $\boldsymbol{W}_{\text{out}}, \boldsymbol{W}_B, \boldsymbol{W}_C$, and $\boldsymbol{W}_{\boldsymbol{\Delta}, \downarrow}, \boldsymbol{W}_{\boldsymbol{\Delta}, \uparrow}$; (ii) $\boldsymbol{W}_{\text{out}}, \boldsymbol{W}_B, \boldsymbol{W}_C$ and $\boldsymbol{W}_{\boldsymbol{\Delta}, \downarrow}$; and (iii) $\boldsymbol{W}_{\text{out}}$ and $\boldsymbol{W}_{\boldsymbol{\Delta}, \uparrow}$. For SDT, we set the channel freeze ratio at 99% across all scenarios. We select the state freeze ratio $\alpha$ from the set $\{75\%, 90\%, 95\%\}$ and apply LoRA exclusively to $\boldsymbol{W}_{\text{out}}$ to maintain a comparable number of trainable parameters. Residual connections and biases are frozen in this experiment. For the warmup, we employ 500 data batches to fully train the SSM modules prior to dimension selection, except for the RTE task in GLUE, where we use 250 batches due to its limited dataset size. Note that the parameters are reverted back after the warmup stage.

**Additional Results on Mamba-II.** For Mamba-II, applying SDT is not straightforward because Mamba-II further constrains $A$ such that all (non-zero) entries must have the same value. Therefore, our original dimension selection approach cannot be directly applied here. We consider a naive extension of SDT by selecting dimensions in the projection matrices for input mapping vector $B$ and the projection matrices for output mapping vector $C$ using their respective magnitude, and fine-tune the selected dimensions and all elements of state transition matrix $A$.

Tables 20 and 21 compare the performance on Mamba-II. The results demonstrate that SDT consistently outperforms LoRA on Mamba-II models.

| Model | Mamba-II-130M | | | Mamba-II-1.3B | | | | |
|---|---|---|---|---|---|---|---|---|
| Dataset | Params (%) | DART | | Params (%) | SAMSum | | | Spider |
| Metric (↑) | | METEOR | BLEU | | R1 | R2 | RL | Acc. |
| LoRA | 0.3354 | 68.71 | 48.09 | 0.1614 | 49.73 | 26.14 | 41.53 | 72.36 |
| LoRA & SDT | 0.3393 | **70.60** | **48.93** | 0.1767 | **50.72** | **27.21** | **42.54** | **84.15** |

*Table 20.* **Performance comparison between SDT and LoRA on Mamba-II-130M and Mamba-II-1.3B. Bold** numbers indicate the best performance for each task.

| Model | Mamba-II-130M | | | | | | |
|---|---|---|---|---|---|---|---|
| Dataset | Params (%) | GLUE | | | | | |
| Accuracy (↑) | | RTE | MRPC | SST2 | QNLI | QQP | MNLI |
| LoRA | 0.3354 | 63.4 | 80.9 | 89.1 | 85.3 | 87.1 | 78.6 |
| LoRA & SDT | 0.3393 | **64.3** | **82.3** | **94.1** | **87.0** | **88.3** | **81.1** |

*Table 21.* **Performance comparison between SDT and LoRA on the GLUE** (Wang et al., 2019) **benchmark using Mamba-II-130M. Bold** numbers indicate the best performance for each task.

**Additional Results on Jamba.** Table 22 shows results for SDT and LoRA on additional datasets. Even though the performance improvement is smaller, our method outperforms pure LoRA in most cases. Mamba layers make up only a small part of Jamba, which is a possible reason for smaller performance gains.

| LinProj | S6 | GLUE | DART | | CelebA | SAMSum | | | Spider |
|---|---|---|---|---|---|---|---|---|---|
| | | Avg. | BLEU | MET. | Acc. | R1 | R2 | RL | Acc. |
| LoRA | LoRA | 65.5 | 52.9 | **73.0** | **88.5** | 56.4 | **33.5** | 47.9 | **90.7** |
| | SDT | **67.7** | **53.1** | **73.0** | 88.4 | **56.5** | **33.5** | **48.0** | 89.8 |

*Table 22.* **Performance comparison between SDT and LoRA on pretrained Jamba models. Bold** numbers indicate the best performance for each task. We use Jamba-Tiny-319M to compare the performance of SDT and LoRA on GLUE (Wang et al., 2019), and CelebA (Liu et al., 2015) benchmarks. For all other datasets, we employ Jamba-Mini-52B. We report only the best setting out of three for each method.

**Additional Results for LoRA+.** We extend our investigation to include LoRA+ (Hayou et al., 2024) with SDT and evaluate its performance against LoRA+ across various datasets on both Mamba-I and Mamba-II. The results, presented in Table 23, show that integrating SDT with LoRA+ enhances its effectiveness and achieves superior performance compared to using LoRA+ alone.

| Model | Mamba-I-130M | | Mamba-II-130M | | Mamba-II-1.3B | | | |
|---|---|---|---|---|---|---|---|---|
| Dataset | **DART** | | **DART** | | **SAMSum** | | | **Spider** |
| Metric (↑) | METEOR | BLEU | METEOR | BLEU | R1 | R2 | RL | Acc. |
| LoRA+ | 70.06 | 50.91 | 69.78 | 49.14 | 49.83 | 26.09 | 41.66 | 73.75 |
| LoRA+ & SDT | **70.58** | **51.93** | **70.48** | **49.99** | **50.81** | **27.19** | **42.4** | **84.22** |

*Table 23.* **Performance comparison between LoRA+ and SDT on Mamba-I and Mamba-II. Bold** numbers indicate the best performance for each task. We test all experiments under various parameter settings (<0.4%) for both LoRA+ and SDT, and report the best values.

