# OpenReview forum: "Parameter-Efficient Fine-Tuning of State Space Models"
_ICML.cc/2025/Conference — ICML 2025 poster_

### Official Review · Reviewer_GPVJ · 2025-02-18

**Overall Recommendation:** 3

**Summary:**

This paper studies the fine-tuning of state-space models, in particular, S4 and S6. Empirical studies on fine-tuning the encoder and the decoder using many different existing tuning mechanisms are shown. Then, an SDT-P fine-tuning strategy of the autoregressive modules is proposed based on pruning and then sparsely fine-tuning a set of latent states. It is also empirically shown that the pruning stage can be pruned. Experiments show that the SDT method works better than the traditional LoRA strategy.

## update after rebuttal
I have raised my score to weak accept to reflect my current position.

**Claims And Evidence:**

I find the claims about the empirical advantages of SDT well-supported by experiments.

**Essential References Not Discussed:**

NA

**Experimental Designs Or Analyses:**

I checked the soundness of the experimental designs and found them appropriate.

**Methods And Evaluation Criteria:**

I find the proposed methods and evaluation criteria appropriate for the problem at hand.

**Other Comments Or Suggestions:**

The comments here are mainly about the presentation of the material and the delivery of the theoretical statements. My current evaluation of "weak reject" is mainly due to the presentation issues. I can guarantee that once the author(s) put efforts into addressing the following comments, pushing up my evaluation would be an easy reach.

1. Right now, the fine-tuning strategies studied and compared in this paper are only referred to without any other details provided. While some of them are popular, readers who are not experts on fine-tuning may find it hard to parse all of them, and consequently Table 1, all at once. It would be useful to add a brief and mathematical introduction to some of them. An introduction in the supplementary material would be otherwise useful.

2. Lemma 2 and Theorem 1 need rephrasing to be mathematically correct. In particular, consider the following questions:
   1. In Lemma 2, the quantity in (5) is just a number and does not depend on the initial model $f_0$. This does not seem to correctly reflect the definition of (5).
   2. In Lemma 2, when saying "the minimum number of tunable parameters," do you mean the precise minimum number (so that we know for sure that if the number goes below it, then we'll not be able to do the job) or an upper bound of the minimum number?
   3. In Theorem 1, $L^*$ and $H^*$ are not defined. I think they come from the target model. In that case, it needs to be clearly stated what the target model is.
   4. In Theorem 1, when saying "selectively fine-tuning," what is the training algorithm being used? Or is it just an algorithm-free statement that considers the "best" way of modifying the tunable parameters?
   5. In Theorem 1, what does "accurately represent" mean?

3. Some minor comments:
   1. Technically, an S4 model refers to a model where $\mathbf{A}$ is the sum of a diagonal matrix plus a rank-one matrix, whereas an S4D model is the one with a diagonal $\mathbf{A}$. The author(s) should be more precise.
   2. Throughout the manuscript $\otimes$ is used for entrywise product. This is a very misleading notation as it is often used for the Kronecker product. Consider using $\odot$ or $\circ$ for the Hadamard product instead.
   3. On line 293-294, please be more precise about what "up to the same permutation" means.

**Other Strengths And Weaknesses:**

Strengths:
   * The problem studied in this paper is important and opportune.
   * The empirical study is comprehensive and convincing.

Weaknesses:
   * While the central ideas of the paper are clear, I do have some comments on the presentation of the material. See the section below.
   * The theoretical statements are on the rough side and many statements need rephrasing. Please see the section below.

**Questions For Authors:**

1. Lemma 1 is about the expressiveness of different ways of doing the fine-tuning. It does not consider, for example, the stability of representations, or more naively, how easy it is to reach the target by only fine-tuning the encoder. Can the author(s) comment on this?
2. Lemma 1 seems to diminish the significance of the proposed method, as it says that fine-tuning can be done without changing the autoregressive units. This may be related to the first point, but can the authors provide more clarification on this in the manuscript so that the transition from section 4 to 5 is smoother?
3. On line 279, the author(s) wrote "assume all hidden dimensions are active." What does that mean precisely?
4. What is the efficiency of the proposed method compared to, for example, LoRA? For example, Figure 2 is mainly a comparison based on the same number of tunable parameters. What about a comparison based on the fine-tuning time? Also, how do these change as the sequence lengths grow?
5. How are $\alpha$ and $\beta$ in Algorithm 1 selected in Table 3 and 4 and why is it a fair comparison to LoRA with whatever the hyperparameters you selected there? These are necessary to be discussed in the main manuscript.

**Relation To Broader Scientific Literature:**

This paper analyzes the fine-tuning of a promising class of large language models, which is certainly an important problem.

**Theoretical Claims:**

I did not check the correctness of the proofs. In fact, I am not satisfied with the presentation of some theoretical statements, which I will clarify in the comments/questions sections.

---

> ### Author Rebuttal · Authors · 2025-04-01
>
> Thank you for your thoughtful feedback. We've addressed all your concerns below.
>
> ---
>
> > Q1: Lemma 2 Clarifications
> > * (i) Use $\odot$ or $\circ$ for Hadamard product?
> > * (ii) Clarify "same permutation"?
> > * (iii) (5) seems independent of $f_0$—is this correct?
> > * (iv) What does "all hidden dimensions are active" mean?
> > * (v) Is it the exact or an upper bound on tunable parameters?
>
> * (i) Fixed.
> * (ii) "Up to the same permutation" means that the same permutation matrix $P$ is applied to all three parameters of the initial model $f_0$. This yields a model that is functionally equivalent but with permuted hidden dimensions, giving
>     $$\Theta_0= \\{(P^\top \overline{A}_0 P, P^\top \overline{B}_0, C_0 P) : P \text{ is a permutation matrix}\\}.$$
>     Here, $\overline{A}_0$, $\overline{B}_0$, and $C_0$ are the parameters of $f_0$ before fine-tuning.
>
> * (iii) The quantity in (5) does, in fact, depend on the initial model $f_0$ through the constraint set $\Theta_0$. To prevent confusion, we rephrased Lemma 2 to highlight this constraint set.
>
> * (iv) It means that (for target model $f_\star$), all hidden dimensions are non-zero.
>
> * (v) When the assumption (iv) holds, it refers to the exact minimum.
>
> **`Preview`** Revised Lemma 2 with improved clarity (https://anonymous.4open.science/r/ce7/l2.png). Notations are from Sec. 5.1.
>
> > Q2: Theorem 1
> > * (i) Define $L^\star$ and $H^\star$—are they from the target model?
> > * (ii) Does "selectively fine-tuning" imply a specific algorithm?
> > * (iii) Clarify "accurately represent."
>
> * (i) $L^\star$ and $H^\star$ are indeed the number of layers and the hidden dimension of the target model $f_\star.$ We have updated statement for a better introduction of the target model and the corresponding notations.
> * (ii) The statement focuses solely on model expressiveness, independent of training algorithms.
> * (iii) It means "functionally equivalent," i.e., the fine-tuned model $f$ satisfies $f(x) = f_\star (x)$ for all input sequences $x$.
>
> **`Preview`** Revised version of Theorem 1 with improved clarity (https://anonymous.4open.science/r/ce7/t1.png).
>
> > Q3: Lemma 1
> > * (i)  Lemma 1 covers only expressiveness, not optimization—please comment.
> > * (ii) Lemma 1 suggests fine-tuning linear projections alone is sufficient; does this diminish your method’s significance?
>
>
> * (i) **`New Exp.`** Lemma 1 analyzes only theoretical expressiveness. Empirical results (https://anonymous.4open.science/r/ce7/f1.png) confirm fine-tuning $W_{\text{in}}$ alone matches tuning $W_B$, $W_C$, $W_{\Delta, \uparrow}$ across three GLUE tasks.
> * (ii) Great point. To clarify, in Lemma 1, $A$ and $W_{\Delta,\downarrow}$ are fixed. While $W_{\text{in}}$'s expressiveness includes that of $W_B$, $W_C$, and $W_{\Delta,\uparrow}$, it does not cover the expressiveness of $A$, which is essential for seq2seq operations. Thus, $W_{\text{in}}$ offers a portion of expressivity for SSM, yet it alone remains insufficient to attain optimal performance.
>
> **`Preview`**  Updated Lemma 1 to emphasize fixed $A$ (https://anonymous.4open.science/r/ce7/l1.png) and will further smooth the Sec. 4–5 transition.
>
> > Q4: Be clear whether the model is S4 or S4D.
>
> We specifically use S4D (diagonal $A$) and will clarify this in Sec. 1, 3, and 5.
>
> > Q5: Fine-tuning strategies lack mathematical details.
>
> Thanks. We'll revise Sec. 3 ("Preliminaries") to include mathematical details on fine-tuning methods.
>
> > Q6: What is the fine-tuning time compared to LoRA?
>
> **`New Exp.`** Based on your feedback, we conducted two additional experiments: (i: https://anonymous.4open.science/r/ce7/f2.png) a performance comparison between SDT and LoRA under varying time budgets in a synthetic setting; and (ii: https://anonymous.4open.science/r/ce7/f3.png) a runtime analysis of SDT and LoRA for training on a single batch using a pretrained model, across different sequence lengths. We observe that our method is slightly more efficient than LoRA, particularly as the sequence length increases, because LoRA introduces additional matrix multiplications, while SDT does not.
>
> > Q7: How were $\alpha$, $\beta$ in Table 3/4 chosen, and why is it a fair comparison to LoRA?
>
> **`New Exp`** We fix $\beta = 0.99$ and sweep $\alpha \in \\{0.75, 0.90, 0.95\\}$. To ensure a fair comparison with LoRA, we use similar parameter budgets and hyperparameter sets of the same size for both methods: 45 configurations per method (15 learning rates × 3 method-specific settings). For LoRA, we additionally compare the three chosen configurations against alternative settings with similar parameter budgets (https://anonymous.4open.science/r/ce7/f4.png), showing that our selected configurations perform reasonably well.
>
> ---
>
> **Final Note:** In addition, we’ll summarize all notations in the appendix. Since you said raising your score would be “an easy reach” once issues were addressed, we hope our response meet that bar. We’d appreciate an updated score and are happy to clarify anything further.

---

### Official Review · Reviewer_TAMW · 2025-03-16

**Overall Recommendation:** 3

**Summary:**

This paper investigates the performance of popular parameter-efficient fine-tuning methods (PEFT) (e.g., LoRA and its variants) when applied to SSMs like Mamba and hybrid models such as Jamba. It finds that LoRA-based methods consistently outperform other PEFT approaches, especially when applied to linear projection matrices, but fail to improve performance significantly when directly applied to SSM modules. To address this, the authors propose Sparse Dimension Tuning (SDT), a specialized PEFT method designed explicitly for fine-tuning SSM modules by selectively training specific channel and state dimensions based on theoretical insights. Extensive experiments demonstrate that combining SDT for SSM modules with LoRA for linear projections achieves state-of-the-art performance across diverse tasks, including natural language understanding, generation, and computer vision benchmarks, confirming SDT's effectiveness and efficiency compared to existing methods.

**Claims And Evidence:**

Yes. The claims are backed by extensive experiments and theoretical analysis.

**Essential References Not Discussed:**

No.

**Experimental Designs Or Analyses:**

The authors benchmark PEFT methods on both synthetic datasets (using deep S4 models) and real-world tasks spanning GLUE, DART, SAMSum, Spider, CIFAR-10, and CelebA with SSM-based (Mamba) and hybrid (Jamba) architectures; while these experiments are well-controlled with careful hyperparameter tuning and fair parameter budget comparisons, a minor issue arise:
- The focus on SSM-based and hybrid models limits the generalizability of the findings, suggesting that broader architectural testing could further strengthen the evidence.

**Methods And Evaluation Criteria:**

The proposed methods and evaluation criteria make sense for the problem. The paper introduces SDT specifically to address the limitations of existing PEFT methods on SSM-based models, and it benchmarks these methods using a diverse set of tasks and datasets—including GLUE, DART, SAMSum, Spider, CIFAR-10, and CelebA—which are well-recognized for assessing performance in both language and vision applications.

**Other Comments Or Suggestions:**

See above.

**Other Strengths And Weaknesses:**

Strengths:
- Creative integration of established PEFT techniques like LoRA with a novel Sparse Dimension Tuning (SDT) method tailored for SSM modules.
- Advances the state-of-the-art in fine-tuning SSM-based and hybrid models, which is valuable for efficient language modeling.

Weakness:
- Limited evaluation on a broader range of model architectures beyond SSM-based and hybrid models, which may affect the generalizability of the findings.

**Questions For Authors:**

See above.

**Relation To Broader Scientific Literature:**

The paper’s contributions are well-integrated with the broader scientific literature by extending established PEFT methods—such as prompt tuning, prefix-tuning, BitFit, and especially low-rank adaptations like LoRA—to the realm of SSMs, which have been previously developed for efficient long-sequence modeling.

**Theoretical Claims:**

The proofs for the main theoretical claims, e.g., Lemma 1, Lemma 2, and Theorem 1 and their logical structures and derivations appear sound under the stated assumptions.

---

> ### Author Rebuttal · Authors · 2025-04-01
>
> We thank the reviewer for the encouraging feedback, particularly for recognizing that (i) our method is creative and rational, (ii) our claims are supported by extensive experiments and theoretical analysis, (iii) our theoretical claims are sound, (iv) we advance SOTA results in fine-tuning SSM-based and hybrid models, and (v) our contributions are well-integrated with the broader scientific literature.
>
>
> ---
>
> > Q: Limited applicability beyond SSM-based and hybrid models.
>
> Our proposed SDT algorithm is intentionally tailored for SSMs, with our theoretical analysis and experiments specifically focused on this model class.
>
> Regarding your concern, while applying SDT to pure Transformers is beyond our current scope, we demonstrate its broader potential through experiments on Mamba-II, whose SSD module closely resembles attention [1]. When applied to SSD modules with LoRA on linear projection matrices, SDT consistently outperforms LoRA alone on Mamba-II across diverse tasks (Sec E.2).
>
> Furthermore, we believe the theoretical insights and proof techniques from our analysis could prove valuable for developing PEFT methods for other architectures.
>
>
> ---
>
> *References:*
>
> [1] Dao, T. and Gu, A. Transformers are SSMs: Generalized models and efficient algorithms through structured state space duality. In International Conference on Machine Learning, 2024.
>
> **Final Note:**  Thank you again for your valuable comments. We are grateful that you appreciate our paper’s contributions in theory, methodology, and experimentation. If you have any remaining questions, please do not hesitate to let us know. Assuming our responses have addressed your concerns satisfactorily, we kindly ask you to consider raising your score and supporting our paper.

---

> > ### Comment · Reviewer_TAMW · 2025-04-06
> >
> > Thank you for the clarification. I will keep my current score.

---

> > > ### Author Response · Authors · 2025-04-07
> > >
> > > We appreciate your engagement during the discussion, and thank you for supporting the acceptance of our paper.

---

### Official Review · Reviewer_biYn · 2025-03-16

**Overall Recommendation:** 2

**Summary:**

This paper investigated the PEFT for SSM, like MAMBA.

**Claims And Evidence:**

yes

**Essential References Not Discussed:**

NO

**Experimental Designs Or Analyses:**

yes

**Methods And Evaluation Criteria:**

yes

**Other Comments Or Suggestions:**

No

**Other Strengths And Weaknesses:**

Strengths:

1. The paper is well written, and there are many PEFT studies for Transformer-based LLM, not many PEFT for SSM.

2. The paper contains many experiemntal result to support their claim in the supplementary materials.

Weaknesses:

1. The proposed PEFT is incremental, such idea is mainly from the PEFT for transformer-based LLM.
2. The theoretical analysis is mainly for S4 or S6, how about S5 and Mamba2?

**Questions For Authors:**

1. The proposed PEFT is incremental, such idea is mainly from the PEFT for transformer-based LLM.
2. The theoretical analysis is mainly for S4 or S6, how about S5 and Mamba2?

**Relation To Broader Scientific Literature:**

it mainly focus on PEFT for SSM.

**Theoretical Claims:**

yes

---

> ### Author Rebuttal · Authors · 2025-04-01
>
> We thank the reviewer for acknowledging that (i) our paper is well written, (ii) it tackles the gap where PEFT studies favor Transformer over SSM, and (iii) it contains many experimental results to support the claims.
>
> ---
>
> > Q1: The proposed PEFT is incremental, such idea is mainly from the PEFT for transformer-based LLM.
>
> We respectfully disagree. SDT is not just an incremental tweak of PEFT for Transformer-based LLMs—it’s built on a fresh theoretical analysis tailored for SSMs’ unique parameter structure. Beyond that, our paper delivers substantial contributions that stand firm, even if the method is viewed as incremental by the reviewer:
> * (i) **(One of the First)** A systematic benchmark of PEFT methods on SSM models,
> * (ii) **(First)** A theoretical analysis of PEFT in the SSM setting, and
> * (iii) **(One of the First)** A new PEFT method designed specifically for SSMs, supported by both theory and strong empirical results.
>
> Together, these contributions offer new insights into PEFT beyond what has been explored in Transformer-based LLMs.
>
>
> > Q2: The theoretical analysis is mainly for S4 or S6, how about S5 and Mamba2?
>
> * **`New Theoretical Results`** **S5**: Although our original analysis primarily focused on S4 and S6, based on your comment, we have extended our theoretical results (Lemma 2 and Theorem 1) to include S5. Specifically, Lemma 2, which characterizes the minimal number of parameters required to update a frozen S4 model for functional equivalence to a target S4, has been extended to S5 with similar conclusions, accounting for multi-channel handling. Additionally, Theorem 1, regarding the expressive power of SDT-P with LoRA on simplified SSM-based models, also holds true for S5. We provide the detailed extension for S5 here: https://anonymous.4open.science/r/ce7/l9.png.
>
>
> * **Mamba-II**: Although extending our theoretical analysis to Mamba-II is non-trivial, we successfully adapt our method to this architecture, as detailed in Sec. C.2 and E.2, and evaluate it on diverse tasks, including GLUE, DART, SAMSum, and Spider. Our method consistently outperforms LoRA alone on Mamba-II, highlighting its generalizability beyond the architectures explicitly covered in our theoretical analysis. We will acknowledge this theoretical limitation explicitly in the conclusion section, emphasizing that extending our analysis to Mamba-II is an important direction for future work.
>
> ---
>
> **Final Note:** Thank you for sharing your concerns. While we understand your perspective, we kindly remind you of the substantial contributions our paper makes and the efforts we've undertaken to address your comments. We would greatly appreciate it if you could reconsider your evaluation score and support the acceptance of our paper.

---

### Official Review · Reviewer_Bhh7 · 2025-03-17

**Overall Recommendation:** 4

**Summary:**

This paper investigates how PEFT methods perform on State Space Models (SSMs) (e.g. the Mamba architecture) and identifies which model components are best to target. It provides a comprehensive benchmark of existing PEFT techniques on SSM-based language models and hybrid architectures with Jamba. A key finding is that LoRA and its variants consistently outperform other PEFT approaches on SSM models​. However, LoRA’s benefit comes mainly from fine-tuning the linear projection matrices (e.g. input/output projections) – applying LoRA to the SSM-specific components does not yield additional gains​. In fact, prompt-based tuning methods (like prefix or prompt tuning) are found largely ineffective on SSMs, essentially only adjusting the initial hidden state (a severely limited form of fine-tuning, as proven in the paper’s Proposition 1)​. These insights highlighted a gap: none of the standard PEFT methods are well-suited for the internal SSM parameters. To address this, the authors propose a new method called Sparse Dimension Tuning (SDT) – a PEFT strategy tailored for SSM layers. SDT works by selectively fine-tuning only a subset of the state-space channels and state dimensions, while freezing or pruning the rest. When combining SDT (for SSM layers) with LoRA (for the linear projection layers), the paper achieves state-of-the-art fine-tuning performance on multiple benchmarks​. The approach matches or surpasses the best existing methods while training only a small fraction of model parameters, validating the effectiveness of the proposed methodology across extensive experiments.

**Claims And Evidence:**

Yes.

**Essential References Not Discussed:**

No.

**Experimental Designs Or Analyses:**

Yes.

**Methods And Evaluation Criteria:**

Yes.

**Other Comments Or Suggestions:**

No.

**Other Strengths And Weaknesses:**

Stength:

1. This is a well executed paper that systematically studied PEFT for the emerging neural architectures, including Mamba, Mamba 2 and its hybrid variant. The empirical results are strong and comprehensive.

**Questions For Authors:**

No.

**Relation To Broader Scientific Literature:**

It is related to Parameter Efficient Fine Tuning, State Space Models and Hybrid State Space Models. No previous works on systematic studies of PEFT on SSMs.

**Theoretical Claims:**

No.

---

> ### Author Rebuttal · Authors · 2025-04-01
>
> We are delighted that the reviewer likes the paper, recognizing it as (i) well-executed, (ii) a systematic study of PEFT for emerging neural architectures (Mamba, Mamba 2, and hybrid variants), and (iii) providing strong and comprehensive empirical results. Thank you for your encouragement!

---

### Decision · Program_Chairs · 2025-05-01

**Decision:**

Accept (poster)

**Comment:**

This work performs a detailed study of parameter-efficient fine-tuning (PEFT) for the recently popular family of state space models (SSM). The contributions include a systematic empirical study of existing PEFT methods, and introducing a new method tailored for SSMs that applies to several variants including S4, Mamba, and more models that the authors included during the reviewer discussion. Reviewers agreed that the problem domain of PEFT for SSMs is important and relatively new, and that the paper's contributions are well-motivated and sound.